# Measurement Manipulation of the Matrix Sensing Problem to Improve Optimization Landscape

## Abstract

This work studies the matrix sensing (MS) problem through the lens of the Restricted Isometry Property (RIP). It has been shown in several recent papers that two different techniques of convex relaxations and local search methods for the MS problem both require the RIP constant to be less than 0.5 while most real-world problems have their RIPs close to 1. The existing literature guarantees a small RIP constant only for sensing operators having an i.i.d. Gaussian distribution, and it is well-known that the MS problem could have a complicated landscape when the RIP is greater than 0.5. In this work, we address this issue and improve the optimization landscape by developing two results. First, we show that any sensing operator with a model not too distant from i.i.d. Gaussian has a slightly higher RIP than i.i.d. Gaussian. Second, we show that if the sensing operator has an arbitrary distribution, it can be modified in such a way that the resulting operator will act as a perturbed Gaussian with a lower RIP constant. Our approach is a preconditioning/mixing technique that replaces each sensing matrix with a weighted sum of all sensing matrices. This approach does not require taking new measurements (which is not possible in many applications) and relies only on mixing existing measurements. We numerically demonstrate that the RIP constants for different distributions can be reduced from almost 1 to less than 0.5 via the preconditioning of the sensing operator.

## 1 Introduction

In this paper, we focus on an important class of problems in non-convex optimization and machine learning, named matrix sensing. The goal of the matrix sensing problem is to recover a low-rank matrix from a set of limited linear measurements. To be more specific, given $m$ sensing matrices $A_1, \ldots, A_m \in \mathbb{R}^{n \times n}$, we define the linear sensing operator $\mathcal{A}$ as $\mathcal{A}(M) = [\langle A_1, M \rangle, \ldots, \langle A_m, M \rangle]^T$ for all $M$. The matrix sensing problem is formulated as the following non-convex optimization problem:

$$\min_{M \in \mathbb{R}^{n \times n}} \frac{1}{2} \|\mathcal{A}(M) - b\|^2 \quad \text{subject to} \quad \text{rank}(M) = r. \tag{1}$$

where $b = \mathcal{A}(M^*)$ is the observed vector, $M^*$ is the unknown ground truth matrix, and $r$ denotes the rank of $M^*$. Since the matrix sensing problem for an arbitrary solution $M^*$ (being a rectangular matrix or a square sign indefinite matrix) can be converted to an expanded matrix sensing problem whose solution is a symmetric and positive semidefinite matrix (Zhang et al., 2021), we assume that $M^*$ is positive semidefinite and symmetric without loss of generality.

The matrix sensing problem has a wide range of real-world applications in signal processing and machine learning, such as the training of neural networks (Li et al., 2018), reconstruction of images and videos (Fowler et al., 2012; Baraniuk et al., 2017), wireless sensor network (Razzaque et al., 2013), and quantum computing (Shabani et al., 2011; Ayanzadeh et al., 2020). It has attracted significant attention in recent years as it sheds light on a broad range of non-convex optimization problems, serving as a theoretical guarantee in deep learning theory (Li et al., 2018; Scarlett et al., 2022). The complexity of the matrix sensing problem lies in the low-rank structure that creates spurious solutions, which makes local search algorithms with a random initialization become stuck at a wrong second-order critical point rather than the ground truth (Chen et al., 2019).

To overcome the above-mentioned non-convexity, one line of research relaxes this problem into a convex semi-definite program (SDP) (Candès & Recht, 2012; Recht et al., 2010), by replacing the rank constraint with a nuclear norm

constraint. However, solving the SDP relaxation requires a large amount of calculations (Candes & Recht, 2013). Another popular way to deal with the low-rank constraint is the Burer-Monteiro (BM) factorization (Burer & Monteiro, 2003), which explicitly factorizes the low-rank matrix $M$ into the form $M = XX^\top$ where $X \in \mathbb{R}^{n \times r}$ (note that this factorization uses the fact that $M^*$ is positive definite and symmetric). Hence, the matrix sensing problem can be formulated as

$$\min_{X \in \mathbb{R}^{n \times r}} \frac{1}{2} \|\mathcal{A}(XX^\top) - b\|^2 \tag{2}$$

With this natural reparametrization, the number of parameters reduces from $O(n^2)$ in $M$ to $O(nr)$ in $X$, where $r$ is usually close to 1. Problem (2) is unconstrained, and therefore simple first-order methods, such as Gradient Descent (GD), can be applied to solve the problem. However, the factorized problem (2) is highly non-convex and $\mathcal{NP}$-hard to solve (Gillis & Glineur, 2011; Ge et al., 2017). There have been extensive studies on the optimization landscape of the matrix sensing problem (Candes & Tao, 2010; Candès & Recht, 2012; Recht et al., 2010; Ge et al., 2017; Zhang et al., 2018), and it turns out that the success of both SDP relaxation and local search methods relies on a condition named Restricted Isometry Property (RIP), which is defined below.

**Definition 1.1** (RIP (Candès & Recht, 2012)). Given a natural number $s$, the linear map $\mathcal{A} : \mathbb{R}^{n \times n} \mapsto \mathbb{R}^m$ is said to satisfy the Restricted Isometry Property (RIP) condition of rank $s$ for a constant $\delta$, denoted as $\delta_s \in [0, 1)$, if the inequality

$$(1 - \delta_s) \|M\|_F^2 \leq \|\mathcal{A}(M)\|^2 \leq (1 + \delta_s) \|M\|_F^2 \tag{3}$$

holds for all matrices $M \in \mathbb{R}^{n \times n}$ satisfying $\text{rank}(M) \leq s$.

Intuitively, the RIP is a condition guaranteeing that linear measurements approximately preserve the Euclidean geometry of low-rank matrices. Specifically, a sensing operator satisfies the RIP if it acts nearly as an isometry on the set of low-rank matrices, ensuring that the distances between these matrices are preserved after measurement. When $\delta_s = 0$, solving the matrix sensing problem is trivial, while $\delta_s$ close to 1 implies a complicated landscape for the matrix sensing problem where the number of local minima could be exponential (Yalçın et al., 2023). Note that the RIP constant is not unique. If $\delta_s$ is an RIP constant, every number greater than $\delta_s$ is also an RIP constant.

The RIP condition is crucial for the success of various recovery algorithms, as it underpins their ability to reconstruct the original matrix accurately from compressed measurements. Started by the convex relaxation approach, Recht et al. (2010) and Candès & Recht (2012) demonstrated that when the RIP constant satisfies the inequality $\delta_{5r} \leq 1/10$, the SDP relaxation is exact, allowing for the exact recovery of the ground truth $M^*$. Later, Bhojanapalli et al. (2016) examined the factorized problem (2) and showed that $\delta_{2r} \leq 1/5$ suffices to guarantee that all second-order critical points for (2) correspond to the ground truth solution. Zhu et al. (2018) further established that $\delta_{4r} \leq 1/5$ is sufficient for the global recovery of the ground truth via a local search method. The recent paper (Zhang et al., 2021) showed that $\delta_{2r} < 1/2$ is the tightest bound for guaranteeing such global properties.

Through the lens of RIP, one can guarantee benign optimization landscape and convergence to global optimality, solving the matrix sensing problem either using convex relaxations such as SDP or using non-convex methods such as the BM factorization with a random initialization. Furthermore, when the RIP constant is small, local search has a linear convergence rate for the factorized problem (2) (Zheng & Lafferty, 2015; Lee & Stöger, 2023). Moreover, strict-saddle property holds if $\delta_{2r} < 1/2$, and this result was developed for general low-rank optimization problems beyond matrix sensing (Bi et al., 2022). While the bound $\delta_{2r} < 1/2$ is sharp, it is not satisfied for most real-world problems except in special cases such as a class of isometric distributions.

**Definition 1.2** (Nearly isometrically distributed (Recht et al., 2010)). Let $\mathcal{A}$ be a random variable that takes values in linear maps from $\mathbb{R}^{n \times n}$ to $\mathbb{R}^m$. We say that $\mathcal{A}$ is nearly isometrically distributed if for all $X \in \mathbb{R}^{n \times n}$ it holds that

$$\mathbf{E}\left[\|\mathcal{A}(X)\|^2\right] = \|X\|_F^2$$

and for all $0 < \epsilon < 1$ we have

$$\mathbf{P}\left(\left|\|\mathcal{A}(X)\|^2 - \|X\|_F^2\right| \geq \epsilon \|X\|_F^2\right) \leq \quad 2\exp\left(-\frac{m}{2}\left(\epsilon^2/2 - \epsilon^3/3\right)\right)$$

and for all $t > 0$ we have

$$\mathbf{P}\left(\sup_{X \neq 0} \frac{\|\mathcal{A}(X)\|}{\|X\|_F} \geq 1 + \sqrt{\frac{n^2}{m}} + t\right) \leq \exp\left(-\gamma m t^2\right)$$

for some constant $\gamma > 0$.

Given $0 < \delta < 1$ and $1 \le r \le m$, it turns out that if $\mathcal{A}$ is nearly isometrically distributed, with high probability, i.e., $\delta_r(\mathcal{A}) \le \delta$ if $m = \Theta(rn/\delta^2)$ (Recht et al., 2010; Candès & Plan, 2011). Independent and identically distributed (i.i.d.) Gaussian entries with variance $1/m$ are nearly isometrically distributed, and the literature of matrix sensing has heavily relied on the i.i.d. and Gaussian assumptions to justify the use of RIP. However, in practice we often have no prior knowledge of the distribution of the sensing matrices, and in addition the independence assumption is hardly satisfied.

There are many applications for which it is not possible to collect measurements whose sensing operators are i.i.d. Gaussian or to increase the number of measurements to reduce the RIP. An example is the power systems state estimation (PSSE) problem where the goal is to learn the electrical signals of a power grid from sensory data. The number of measurements cannot go beyond the number of lines and nodes in the network and each measurement matrix has a structure conforming with the network topology. Motivated by such applications for which there is no flexibility in collecting measurements with favorable properties, the objective of this paper is to study how a given set of measurements can be manipulated to improve the RIP. To this end, we first study the case where the problem is not Gaussian due to small perturbations, and we derive an upper bound on the change to the RIP constant in terms of the distance of the distribution of the given operator from a Gaussian distribution. Next, we study whether an operator with an arbitrary distribution can be modified so that it acts as a perturbed Gaussian for which the above result on its RIP constant can be applied. For the case where the true distribution deviates significantly from normal distributions, we introduce a preconditioning algorithm that replaces each sensing matrix with a weighted sum of all sensing matrices. We discuss how this technique makes the resulting operator behave similarly to perturbed Gaussian distributions, leading to a reduction in the RIP constant and improving the optimization landscape. Note that our preconditioing technique mixes existing measurements and does not require obtaining new measurements.

While our focus is on low-rank matrix sensing, similar ideas have been explored in compressed sensing. For example, (Wang & Qu, 2017) propose an SVD-based weighted measurement matrix to improve restricted isometry constants (RICs) in sparse recovery, and (Herman & Strohmer, 2010) analyze general perturbations in sensing matrices and their effect on signal recovery. These works illustrate the benefits of preconditioning and perturbation analysis for recovery guarantees. Our contribution extends these ideas to low-rank matrix sensing, addressing a new problem: improving the optimization landscape when the sensing matrices cannot be changed.

The paper is organized as follows. In Section 2, we illustrate the high-level idea of this work through a practical application. In Section 3, we demonstrate the robustness of the RIP constant to small perturbations to the sensing operator. We show that nearly-isometric measurements under a modest perturbation continue to satisfy the RIP, thereby ensuring the reliable recovery of low-rank matrices. This finding is significant as it broadens the applicability of matrix sensing techniques to real-world scenarios by relaxing the restrictive Gaussian assumption.

Furthermore, we investigate the role of orthogonalization in enhancing the optimization landscape of the matrix sensing problem. In Section 4, we show that the orthogonalization of the sensing matrices can improve the RIP constant, making the landscape more favorable for efficient recovery algorithms. To achieve this, we propose a novel preconditioning method that optimizes the mixing of the measurements to reduce the RIP constant. We provide a theoretical analysis for the proposed method, and empirically show that it is highly effective on various types of measurement distributions, including Poisson, uniform, and correlated Gaussian distributions. In particular, we demonstrate that the original RIP constants for these distributions could be close to 1 for which the SDP relaxation and local search methods would fail to work, while the preconditioning technique reduces the RIP to less than 0.5 so that both of these optimization methods can correctly solve the modified problem.

By addressing the above two aspects, our work contributes to a deeper understanding of the matrix sensing problem with non-Gaussian models. We propose practical solutions to enhance recovery performance, paving the way for more robust and efficient applications in matrix sensing and beyond.

**Definitions and Notations** The symbol $\|v\|$ denotes the Euclidean norm of a vector $v$. $\|X\|_F$ denotes the Frobenius norm of a matrix $X$. $\|X\|_M = \max_{i,j} |X_{ij}|$ denotes the largest absolute entry of a matrix $X$. $\|\mathcal{A}\|_\infty = \max_k \max_{i,j} |A_k{}^{ij}|$ denotes the largest absolute entry of a sensing operator $\mathcal{A}$, where $A_k^{ij}$ denotes the $(i,j)$ entry of the matrix $A_k$. $\sigma_i(X)$ denotes the $i$-th largest singular value of a matrix $X$. $\lambda_i(X)$ denotes the $i$-th largest eigenvalue of a symmetric matrix $X$. $\langle A, B \rangle$ is defined as the inner product $\text{tr}(A^T B)$ for two matrices $A$ and $B$ of the same size, where $\text{tr}$ stands for trace. $\mathbf{E}(x)$ denotes the expectation of a random variable $x$. $\mathbf{P}(E)$ denotes the probability of en event $E$. $f = \Theta(g)$ denotes that there exist constants $c_1, c_2 > 0$ such that $c_1 g \le f \le c_2 g$. $f = \mathcal{O}(g)$ denotes that

there exists a constant $c > 0$ such that $f \leq cg$. For a matrix $X$, $\text{vec}(X)$ is the usual vectorization operation by stacking the columns of the matrix $X$ into a vector and $\text{mat}(\cdot)$ is the inverse operator. $\text{VStack}(\cdot)$ denotes concatenating the rows of a matrix into a vector. $[n]$ denotes the integer set $\{1, \ldots, n\}$. $\delta_s(\mathcal{A})$ denotes the smallest value for $\delta_s$ that satisfies the RIP condition of rank $s$ for the sensing operator $\mathcal{A}$. The matrix orthogonality and the orthonormal basis are defined under the standard inner product $\langle \cdot, \cdot \rangle$.

## 2    Illustrative Example

We illustrate the main idea of this work through a real-world application. The low-rank matrix sensing problem studied in this paper naturally appears in power systems, where PSSE is solved every 5 minutes by power system operators Jin et al. (2020). A power system is a graph with $n$ nodes and a set of edges $\mathcal{E}$. Each node of the system has a voltage parameter $x_i$ to be learned. Each measurement $j$ of the network is in the form of

$$b_j = \sum_{i:(j,i)\in\mathcal{E}} \frac{x_j(x_j - x_i)}{z_{ji}}$$

where $z_{ij}$ is a known line parameter and $\frac{x_j(x_j - x_i)}{z_{ji}}$ is the power flown over line $(j, i)$. The right-hand side of the measurement $j$ can be written as

$$b_j = \langle A_j, xx^T \rangle$$

for some matrix $A_i$ that depends on the parameters $z_{ji}$ and the topology of the graph (note that $x$ is the vector of all nodal voltages). We cannot change any measurement model $A_j$ directly. Changing an entry of $A_j$ means removing/adding lines to a physical power grid or changing the reactances of the transmission lines on the streets, which is impossible (the goal is to learn the voltages from the data given by the sensors rather than changing the infrastructure). The existing methods requiring $A_j$ to be Gaussian are not applicable at all since $A_j$ is heavily structured for power systems. We propose the following idea:

- We start with the sensors $1, 2, \ldots, m$ returning the measurements $b_1, \ldots, b_n$.

- We design some coefficient $P_{11}, \ldots, P_{1m}$, and create a mixed measurement $P_{11}b_1 + \cdots + P_{1m}b_m$. We replace measurement 1 with this new combined measurement. Then, the new measurement can be written as $\langle \tilde{A}_1, xx^T \rangle$, where $\tilde{A}_1$ is equal to $P_{11}A_1 + \cdots + P_{1m}A_m$.

- Note that the mixing idea cannot generate arbitrary values for $\tilde{A}_1$. For example, if there is no physical line between nodes 2 and 3, then the (2,3) entry of all matrices $A_1, \ldots, A_m$ are zero and so the (2,3) entry of $\tilde{A}_1$ is also zero no matter how we select the coefficients $P_{1j}$'s.

- We then proceed and replace measurement 2 with a new mixed measurement $P_{21}b_1 + \cdots + P_{2m}b_m$. We proceed with the replacement of all measurements similarly.

- Using this idea, we exploit the existing measurements/sensors, and do not require new measurements that are not physically infeasible. The question is: How can $P_{ij}$'s be designed so that the process of learning $x$ becomes simpler?

To explain the above idea in a general context, the above pre-conditioning technique to be studied in this paper allows us to make linear combinations of the original sensing matrices, and we cannot change $A$ to arbitrary $\tilde{A}$ such as a Gaussian i.i.d. sensing operator. We use mixed measurements to obtain

$$\tilde{A}_i = \sum_{j=1}^{m} P_{ij}A_j, \quad \forall i \in \{1, \ldots, m\}$$

The new sensing operator can only be in the linear span space of the original sensing matrices.

## 3 Perturbed Isometrical distribution

In this section, we investigate the behavior of RIP under deviations from the standard i.i.d. Gaussian assumption. The existing literature establishes that if RIP is below 0.5, the Matrix Sensing problem is easy, and for i.i.d. Gaussian samples, RIP can decrease below 0.5 as the sample size grows. However, for non-Gaussian distributions, RIP may remain above 0.5 even with infinitely many samples, as RIP may not vary smoothly with changes in measurement distributions. We aim to show that if the deviation from Gaussian is modest, increasing the number of samples can still reduce RIP below 0.5, aligning it with Gaussian-like behavior. This is significant because for distributions far from Gaussian, RIP may remain large despite an infinite number of samples, but our result shows that moderate deviations still allow for improvement. We prove that given an arbitrary sensing operator $\mathcal{A}$, if $\mathcal{A}$ is perturbed via another operator that is bounded by $\varepsilon$, then its RIP constant will be increased by at most $\mathcal{O}(mn^2\varepsilon)$.

**Theorem 3.1.** *Consider an arbitrary operator $\mathcal{A}$ with the RIP constant $\delta_s \in [0,1)$. Let $\varepsilon$ be a nonnegative constant such that $\varepsilon < \frac{1-\delta_s}{2mn^2\|\mathcal{A}\|_\infty}$. For every bounded perturbation operator $\mathcal{N}$ with $\|\mathcal{N}\|_\infty \leq \varepsilon$, the perturbed sensing operator $\mathcal{A} + \mathcal{N}$ satisfies the RIP condition of rank $s$ with the constant $\delta_s + (4mn^2\|\mathcal{A}\|_\infty\varepsilon + mn^2\varepsilon^2(1-\delta))/(2 + mn^2\varepsilon^2)$.*

See Appendix A for proof.

If $\mathcal{N}$ is chosen as $-\mathcal{A}$, then the RIP condition is not satisfied. Similarly, if $N_1, \ldots, N_m$ are chosen in a way that the $(i,j)$ entries of all matrices $A_1 + N_1, \ldots, A_m + N_m$ are zero for some indices $i$ and $j$, then the RIP condition again no longer holds. For these reasons, the existence of an upper bound on $\varepsilon$ in Theorem 3.1 is necessary.

*Remark* 3.2. With series expansion at $\varepsilon = 0$, the RIP constant derived in Theorem 3.1 can be approximated by $\delta_s + mn^2 \left(2\|\mathcal{A}\|_\infty\varepsilon + \frac{1}{2}(1-\delta_s)\varepsilon^2 - \|\mathcal{A}\|_\infty mn^2\varepsilon^3 + \mathcal{O}(\varepsilon^4)\right)$. On the other hand, since $\mathcal{A}$ satisfies the RIP condition with the constant $\delta_s$, the term $\|\mathcal{A}\|_\infty$ can be bounded by choosing a matrix $X$ whose entry at the position of the largest element of $\mathcal{A}$ is 1 and whose remaining entries are 0. Hence, $\|X\|_F^2 = 1$ and $\|\mathcal{A}\|_\infty^2 \leq \sum_{i=1}^m \langle A_i, X\rangle^2 \leq (1+\delta_s)\|X\|_F^2$, indicating that $\|\mathcal{A}\|_\infty \leq \sqrt{1+\delta_s}$. Thus, the RIP condition for $\mathcal{A} + \mathcal{N}$ can be upper bounded by $\delta_s + mn^2\varepsilon[2(1+\delta_s)^{1/2} + \frac{1}{2}(1-\delta_s)\varepsilon]$ up to the first-order approximation. If we apply the upper bound of $\varepsilon$ to our result, our upper bound on RIP due to first-order approximation is

$$\delta_s + \frac{(1-\delta_s)(1+\delta_s)^{1/2}}{\|\mathcal{A}\|_\infty} + \frac{1-\delta_s}{8mn^2\|\mathcal{A}\|_\infty^2}$$

Theorem 3.1 studies bounded perturbation operators $\mathcal{N}$ in the worst case. We will improve the results by relaxing the boundedness of the perturbation.

**Corollary 3.3.** *Consider an arbitrary operator $\mathcal{A}$ with the RIP constant $\delta_s \in [0,1)$. Consider also a perturbation operator $\mathcal{N}$ such that $\|\mathcal{N}\|_\infty$ is sub-Gaussian with mean 0 and variance proxy $\sigma^2/m$. For every $c > 0$ and $\sigma < \frac{1-\delta_s}{2c\sqrt{m}n^2\|\mathcal{A}\|_\infty}$, with probability at least $1 - 2\exp(-c^2)$, the operator $\mathcal{A} + \mathcal{N}$ satisfies the RIP condition with the constant $\delta_s + c\sqrt{m}n^2\sigma[2(1+\delta_s)^{1/2} + \frac{c}{2\sqrt{m}}(1-\delta_s)\sigma]$.*

See Appendix B for proof.

Building on Corollary 3.3, we refine the RIP bound for a nearly isometrically distributed operator $\mathcal{A}$.

**Theorem 3.4.** *Assume that $\mathcal{A}$ is nearly isometrically distributed and $\|\mathcal{N}\|_\infty$ is sub-Gaussian with mean 0 and variance proxy $\sigma^2/m$. There exist positive constants $c_1$ and $c_2$, independent of the parameters of $\mathcal{N}$ (such as $\sigma$) such that for every $c > 0$ and $\sigma < \frac{1-\delta_s}{2c\sqrt{m}n^2\|\mathcal{A}\|_\infty}$, with probability at least $1 - 2\exp(-c^2) - \exp(-c_1m)$, the operator $\mathcal{A} + \mathcal{N}$ satisfies the RIP condition with the constant $c_2\sqrt{ns\log n/m} + c\sqrt{m}n^2\sigma[2(1+\delta_s)^{1/2} + \frac{c}{2\sqrt{m}}(1-\delta_s)\sigma]$.*

See Appendix C for proof.

*Remark* 3.5. Due to Theorem 3.4, the RIP constant of the perturbed operator $\mathcal{A} + \mathcal{N}$ compared to the RIP of $\mathcal{A}$ has increased from $\mathcal{O}(1/\sqrt{m})$ to $\mathcal{O}(1/\sqrt{m}) + \mathcal{O}(\sqrt{m}\sigma) + \mathcal{O}(\sigma^2)$. Thus, when the perturbation $\sigma$ is small, one can compensate for the influence of the perturbation on the RIP constant by slightly increasing the number of measurements $m$, which will reduce the RIP constant of the perturbed operator to the RIP constant of the unperturbed operator $\mathcal{A}$. This formula shows how many additional measurements are needed to nullify the effect of deviation from a Gaussian distribution.

To summarize above, Theorem 3.1 provides an RIP bound for a fixed sensing operator $\mathcal{A}$ and a bounded perturbation $\mathcal{N}$, and Corollary 3.3 extends this result to a random perturbation $\mathcal{N}$. In Theorem 3.4, we further derive a high probability bound for any nearly isometric random distributed sensing operator $\mathcal{A}$, and prove that the impact of a small perturbation on RIP is small and that increasing the number of measurements $m$ on a small scale can compensate for the increase in RIP.

Additionally, We provide the following theorem on the robustness of RIP under distributional shifts. It formalizes how RIP constants behave under small Wasserstein-1 perturbations and demonstrates stability.

**Definition 3.6** (Wasserstein-1 Distance). Let $\mu, \nu$ be two probability measures on $\mathbb{R}^d$ with finite first moments. The *Wasserstein-1 distance* between $\mu$ and $\nu$ is defined as

$$W_1(\mu, \nu) := \inf_{\pi \in \Pi(\mu, \nu)} \int_{\mathbb{R}^d \times \mathbb{R}^d} \|x - y\|_2 \, d\pi(x, y),$$

where $\Pi(\mu, \nu)$ denotes the set of all couplings of $\mu$ and $\nu$.

**Theorem 3.7** (RIP under Wasserstein perturbation). *Fix $\delta \in (0, 1)$. Suppose $\mathcal{A} \sim P$ is nearly isometrically distributed and achieves RIP with probability $1 - \exp(-c_1 m)$ as $m \geq c_0\, nr \log(n)$. Let $\widetilde{\mathcal{A}} \sim Q$ with $W_1(P, Q) = w$. Then, for $M := 1 + \sqrt{\frac{n^2}{m}}$, we have*

$$\mathbf{P}_Q\left(\delta_r(\widetilde{\mathcal{A}}) \leq \delta\right) \geq 1 - \exp(-c_1 m) - \frac{c_2 M}{\delta}\, w,$$

See Appendix D for proof.

*Remark* 3.8. The RIP for $Q$ holds with asymptotically the same measurement order as for $P$, with an additional failure probability term proportional to $w$. If $w \lesssim \delta/M$, no extra samples or a modest increase in $m$ are needed to keep the same confidence level.

## 4   Preconditioning of Matrix Sensing

In the previous section, we proved that small deviations from nearly isometrically distributed sensing matrices will slightly increase the RIP constant. However, real-world sensing matrices often have unknown probability distributions that cannot be approximated by Gaussian models, for which several empirical results have shown that the RIP constant is often close to 1. To address this issue, we consider a sensing operator $\mathcal{A}$ coming from an arbitrary probability distribution and develop a preconditioning algorithm to improve its RIP constant and make it act as a perturbed Gaussian. Note that our preconditioning technique only mixes existing measurements and cannot arbitrarily change the sensing matrices.

It has been proved in Ma et al. (2023; 2024) that the RIP constant can be reduced if the optimization complexity of the matrix sensing problem is increased, e.g., via a tensor-based lifting technique. However, this incurs a high computational cost and is not applicable to large-scale matrix sensing problems. To avoid this computational complexity, we propose a simple and scalable linear preconditioning method, which replaces every sensing matrix with a linear combination of all the original sensing matrices. More precisely, consider a weight matrix $P \in \mathbb{R}^{m \times m}$ with its $(i, j)$ entry denoted as $P_{ij}$. We construct a preconditioned operator $\tilde{\mathcal{A}}$ with the components $\tilde{A}_1, ..., \tilde{A}_m$ as follows:

$$\tilde{A}_i = \sum_{j=1}^m P_{ij} A_j, \quad \forall i \in \{1, ..., m\}$$

Therefore, $\forall i \in \{1, \ldots, m\}, \forall X$, we have

$$\langle \tilde{A}_i, X \rangle = \sum_{j=1}^m P_{ij} \langle A_j, X \rangle = \sum_{j=1}^m P_{ij} b_j.$$

Hence, $\tilde{\mathcal{A}}(X) = Pb$. The preconditioning is independent of the optimization method (such as local search or convex relaxation) to be used to solve the matrix sensing problem, and the goal of preconditioning is to create a better structure for the sensing operator and thus a better RIP constant. In what follows, we will develop a simple method for designing $P$ and study its impact on the RIP constant.

### 4.1 Orthonormal Bases as Sensing Matrices

The following lemma for Haar distribution is the basis of our method.

**Lemma 4.1** (Instance of the JL lemma (Johnson et al., 1984)). *Let $\{x_j\}_{j=1}^n \subseteq \mathbb{R}^d$, and let $P$ be a $k \times d$ random matrix, consisting of the first $k$ rows of a Haar-distributed random matrix in the orthogonal group $\mathbb{O}(d)$. Given $\epsilon > 0$ and $k = \frac{a \log(n)}{\epsilon^2}$, there are absolute constants $c$ and $C$ such that with probability at least $1 - Cn^{2 - \frac{ac}{4}}$ the inequalities*

$$(1 - \epsilon) \|x_i - x_j\|^2 \leq \binom{d}{k} \|Px_i - Px_j\|^2 \leq (1 + \epsilon) \|x_i - x_j\|^2$$

*hold for all $i, j \in \{1, \ldots, n\}$.*

The orthonormal vectors from the unitary matrix in QR decomposition of i.i.d. Gaussian matrices follow a Haar distribution (Mezzadri, 2007). Given that those orthonormal bases maintain the distances during projection, we are inspired to transform our original sensing operator $\mathcal{A}$ into a preconditioned operator $\tilde{\mathcal{A}}$ with orthonormal bases as vectorized sensing matrices. To be more specific, we first write the sensing operator $\mathcal{A}$ into the vectorized form

$$\mathbf{A} = [\text{vec}(A_1), \text{vec}(A_2), \ldots, \text{vec}(A_m)]^T \in \mathbb{R}^{m \times n^2}.$$

Then, since the inner product of two matrices can be defined as a vector product, it holds that

$$\mathbf{A} \, \text{vec}(X) = \mathcal{A}(X), \quad \forall X \in \mathbb{R}^{n \times n}.$$

By pre-multiplying the above equation with a weight matrix $P \in \mathbb{R}^{m \times m}$, we ideally intend to make the rows of $P\mathbf{A}$ normalized and orthogonal to each other. Since the individual entries of a random orthogonal matrix are approximately Gaussian for large matrices (Meckes, 2019), as $m$ increases, these preconditioned operators are likely to act as i.i.d. Gaussian.

Define the $s$-sparse set $\text{span}_s(\mathcal{A})$ as the set of all matrices $X$ that can be written as $X = \sum_{i=1}^m \alpha_i A_i$ for some coefficients $\alpha_1, ..., \alpha_m$ such that at most $s$ coefficients are nonzero. We say that $A_1, ..., A_m$ are orthonormal if $\langle A_i, A_j \rangle = 0, \forall i \neq j$ and $\langle A_i, A_i \rangle = 1$ otherwise.

**Theorem 4.2.** *Assume that $A_1, ..., A_m$ are orthogonal. It holds that*

$$\frac{\|\mathcal{A}(X)\|^2}{\|X\|_F^2} = 1, \quad \forall X \in \text{span}_s(\mathcal{A})$$

See Appendix E for proof.

Theorem 4.2 is a generalization of Parseval's identity. The set $\text{span}_s(\mathcal{A})$ includes matrices that can be written as the sum of at most $s$ matrices from the set $\{A_1, ..., A_m\}$. As $m$ increases, if this set continues to include orthonormal matrices, the set $\text{span}_s(\mathcal{A})$ grows until it completely covers the low-rank set $\{X \mid \text{rank}(X) \leq s\}$. Thus, it follows from Theorem 3 that as $m$ grows, the RIP constant for orthonormal matrices approaches zero (note that RIP is about taking the minimum and maximum of the ratio $\|\mathcal{A}(X)\|^2/\|X\|_F$ over matrices of rank at most $s$). Hence, Theorem 4.2 justifies the conversion of arbitrary sensing matrices into orthogonal matrices.

### 4.2 Preconditioning Algorithm

Based on the idea of using orthonormal bases as sensing matrices, we propose Algorithm 1, which applies the thin singular value decomposition (SVD) to extract unitary sensing matrices from the given sensing operator.

---

**Algorithm 1** Preconditioned Matrix Sensing

---

1: **for** iteration $= 1, 2, \ldots, m$ **do**
2: $\quad a_i \leftarrow \mathrm{vec}(A_i)$
3: **end for**
4: $U, S, V^\top \leftarrow \mathrm{SVD}(\mathrm{VStack}(a_1, a_2, \ldots, a_m))$
5: **for** iteration $= 1, 2, \ldots, m$ **do**
6: $\quad \tilde{A}_i \leftarrow \mathrm{mat}(V_i^\top)$
7: **end for**
8: **Return** $\tilde{\mathcal{A}} = [\tilde{A}_1, \tilde{A}_2, \ldots, \tilde{A_m}]$

---

*Remark* 4.3. The thin singular value decomposition of $\mathbf{A}$ written as $USV^\top$ will obtain a row-orthogonal matrix $V^\top \in \mathbb{R}^{m \times n^2}$ whose rows are the eigenvectors of $\mathbf{A}^\top \mathbf{A}$. The new sensing matrices $\tilde{A}_i$ obtained by reshaping the rows of $V^\top$ into matrices are perpendicular to each other. For $S = \mathrm{diag}([\sigma_1(\mathbf{A}), \ldots, \sigma_m(\mathbf{A})]) \in \mathbb{R}^{m \times m}$, we could assume $\mathbf{A}$ to be full rank in practice, and since $\sigma_m(\mathbf{A}) > 0$, $S$ becomes invertible. Since the extraction step can be considered as a linear transformation, we can easily calculate the corresponding vector $b' = S^{-1} U^\top b$, and the weight matrix is $P = S^{-1} U^\top$.

The intuition behind the pre-conditioning algorithm is that after pre-conditioning, the individual entries of the new sensing matrix are approximately Gaussian, and hence these preconditioned operators are likely to act as i.i.d. Gaussian with small perturbation. As long as the new upper bound after perturbation is smaller than 0.5, we can obtain favorable properties such as global optimality for the matrix sensing problem.

**Theorem 4.4** (Approximate Gaussianization via SVD Preconditioning). *Let $A \sim \mathcal{N}\left(0, \frac{1}{m}\right) \in \mathbb{R}^{m \times n^2}$ be i.i.d. Gaussian ($m < n^2$), and let $B \in \mathbb{R}^{m \times n^2}$ be any matrix such that $W_1(\mathcal{L}(A), \mathcal{L}(B)) = w$. Let $A = U_A \Sigma_A V_A^\top$, $B = U_B \Sigma_B V_B^\top$ be the thin SVD of $A$ and $B$, and define the preconditioned matrices $\widetilde{B} := V_B^\top \in \mathbb{R}^{m \times n^2}$, $\widetilde{A} := V_A^\top$. Then, in the full-rank regime and $m \ll n^2$, the matrix $\widetilde{B}$ has orthonormal rows, $\widetilde{B}\widetilde{B}^\top = I_m$, and with high probability, its law is close to the Haar measure on the Stiefel manifold in the sense that $W_1\left(\mathcal{L}(\widetilde{B}), \mathcal{L}(\widetilde{A})\right) \lesssim \frac{\sqrt{m}}{n} w$, and the entries of $\widetilde{B}$ are approximately Gaussian, up to deviations vanishing as $n \to \infty$.*

See Appendix F for proof.

*Remark* 4.5. Theorem 4.4 shows that applying thin SVD preconditioning to an arbitrary matrix $B$ produces a matrix $\widetilde{B} = V_B^\top$ with orthonormal rows that is close in Wasserstein distance to Haar-distributed $\widetilde{A} = V_A^\top$. In the tall-and-skinny regime ($m \ll n^2$), the entries of $\widetilde{B}$ become approximately Gaussian. This implies that, regardless of the original distribution of $B$, preconditioning effectively randomizes its rows and contracts the distribution towards Haar/Gaussian. Consequently, the near-isometry properties of $\widetilde{B}$ are enhanced, making it more likely to satisfy RIP conditions and improving the theoretical guarantees for low-rank matrix recovery.

**Theorem 4.6.** *Consider an arbitrary operator $A$ with the RIP constant $\delta_s \in [0, 1)$. Then, the conditioned operator $\tilde{\mathcal{A}}$ also satisfies the RIP condition with the constant $1 - \frac{1 - \delta_s}{\sigma_1^2(\mathbf{A})}$.*

See Appendix G for proof.

**Assumption 4.7.** Assume that singular values of the matrix $\mathbf{A} \in \mathbb{R}^{m \times n^2}$ with $m < n^2$ satisfy

$$\Pr\left\{ \sqrt{n^2/m}(1 - \epsilon) - 1 \le \sigma_i(\mathbf{A}) \le 1 + \sqrt{n^2/m}(1 + \epsilon), \right.$$
$$\left. i \in [m] \right\} \ge 1 - 2\exp\left(-n^2 \epsilon^2 / 2\right), \quad \forall \epsilon > 0.$$

**Assumption 4.8.** Consider two constants $\epsilon$ and $\delta$ such that

$$0 < \epsilon < 1 - \sqrt{\frac{m}{n^2}}, \quad \frac{[1 + \sqrt{\frac{m}{n^2}}(1 + \epsilon)]^2 - 1}{2[1 + \sqrt{\frac{m}{n^2}}(1 + \epsilon)]^2 - 1} < \delta < \frac{1}{2}.$$

**Theorem 4.9.** *Let $\mathcal{A}$ be a nearly isometrically distributed operator. Under Assumption 4.7 and Assumption 4.8, there exist positive constants $c_0$ and $c_1$ depending only on $\delta_s$ such that, with probability at least $1 - \exp(-c_1 m) - 2\exp(-n^2\epsilon^2/2)$, as long as $m \geq c_2 sn \log(n)$, the original sensing operator satisfies $\delta_s(\mathcal{A}) \leq \delta$ and the conditioned sensing operator satisfies $\delta_s(\tilde{\mathcal{A}}) \leq 1 - (1-\delta)/[1 + \sqrt{\frac{n^2}{m}}(1+\epsilon)]^2$.*

See Appendix H for proof.

To shed light on the two assumptions used in Theorem 4.9, note that Gaussian random matrices satisfy Assumption 4.7 as an example. Regarding Assumption 4.8, when $\epsilon \to 0, \delta \to \frac{1}{2}$, and the lower bound of $\delta$ is always smaller than $\frac{1}{2}$. As a result, such pair $(\epsilon, \delta)$ satisfying Assumption 4.8 always exists.

*Remark* 4.10. As $m, n \to \infty$, in the order of $m \gtrsim ns \log n$, it follows from the above theorem that the RIP of the perconditioned operator is similar to that of the original operator. This is important since nearly isometrically distributed operators have small RIPs when $m$ is large and our result says that preconditioning does not transform such optimal operators to sub-optimal operators. In summary, we have shown that preconditioning improves those operators far from nearly isometrically distributed and does not deteriorate the RIP when the original operator is already nearly isometrically distributed. operator. Since Gaussian random matrices satisfy the concentration inequality of the singular values naturally, we can simplify the result of Theorem 4.9 below.

**Corollary 4.11.** *Let $A_1, \ldots, A_m$ be i.i.d. Gaussian random matrices of mean zero and variance $\frac{1}{m}$. Under Assumption 4.8, there exist positive constants $c_0$ and $c_1$ such that, with probability at least $1 - \exp(-c_1 m) - 2\exp(-n^2\epsilon^2/2)$, as long as $m \geq c_0 s(m + n^2 \log(mn^2))$, , it holds that $\delta_s(\mathcal{A}) \leq \delta$ and $\delta_s(\tilde{\mathcal{A}}) \leq 1 - (1-\delta)/[1 + \sqrt{\frac{n^2}{m}}(1+\epsilon)]^2$.*

### 4.3 Simulation Experiments

In this subsection, we will demonstrate the performance of the preconditioning Algorithm 1 for $s = 2r$ since $\delta_{2r}$ determines whether or not SDP relaxations or local search methods succeed to solve the matrix sensing problem. However, measuring the true RIP value $\delta_{2r}$ for any given sensing operator $\mathcal{A}$ requires checking the inequalities (3) for all low-rank matrices $X$ of rank at most $2r$ and determining the maximum and minimum possible values of $\|\mathcal{A}(X)\|_2^2/\|X\|_F^2$ over all rank-$2r$ matrices. This is equivalent to solving a non-convex optimization problem, which is known to be $\mathcal{NP}-$hard. Hence, we will instead measure the empirical RIP constant in our experiments. By

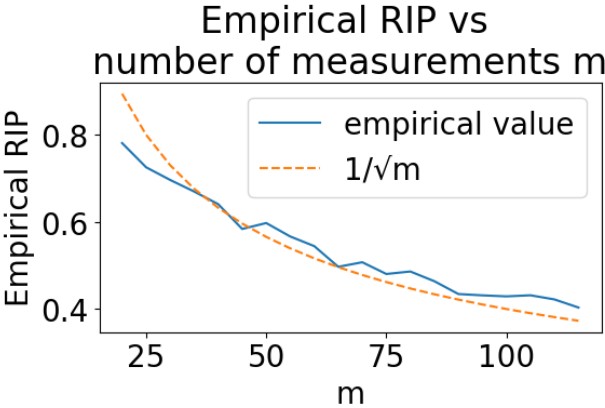

Figure 1: Empirical RIP curve

randomly selecting 1000 Gaussian distributed matrices $M \in \mathbb{R}^{n \times 2r}$ (we simply choose $r = 1$ in the following experiments), we generate 1000 rank-$2r$ matrices $X = M^\top M \in \mathbb{R}^{n \times n}$ to be rank-$2r$ matrices. Afterwards, we calculate $\|\mathcal{A}(M)\|_2^2/\|X\|_F^2$ for all those $X$ matrices and compute the smallest and the largest values, denoted as $\alpha$ and $\beta$ correspondingly. Hence, we obtain the following inequalities over the generated samples of rank-$2r$ matrices:

$$\alpha\|X\|_F^2 \leq \|\mathcal{A}(X)\|^2 \leq \beta\|X\|_F^2. \tag{4}$$

Since rescaling (multiplying the sensing operator $\mathcal{A}$ by a constant $c$) will not affect the landscape of the matrix sensing problem, we multiply all of the above inequalities by $\frac{2}{\alpha+\beta}$ and calculate the empirical RIP constant for $\frac{2}{\alpha+\beta}\mathcal{A}$, which

is $\frac{\beta+\alpha}{\beta-\alpha}$. Given that the set of simulated $X$ is a subset of all low-rank matrices, the simulated RIP is a lower bound for true RIP value. We can see in Figure 1 that for Gaussian distributed sensing matrices, the empirical RIP value decreases as the number of measurements $m$ increases. The empirical RIP curve matches the $m^{-1/2}$ curve, which is the result of the true RIP bound in Recht et al. (2010). Hence, we could treat the empirical RIP value as an accurate measure of the true RIP constant.

### 4.3.1  Synthetic Data

We randomly generate $m$ sensing matrices under different distributions, including nearly isometric distributions such as Gaussian and non-isometric distributions such as Poisson. Besides, we also generate $\mathcal{A}$ with special structures, including low-rank structures and sparse structures. We numerically calculate the empirical RIP value before and after the preconditioning step. We run experiments under different scenarios from $n = 10$ to $n = 50$, and run 100 trials for each scenario to obtain the average empirical RIP value. The results are plotted in Figure 2. We can see from the

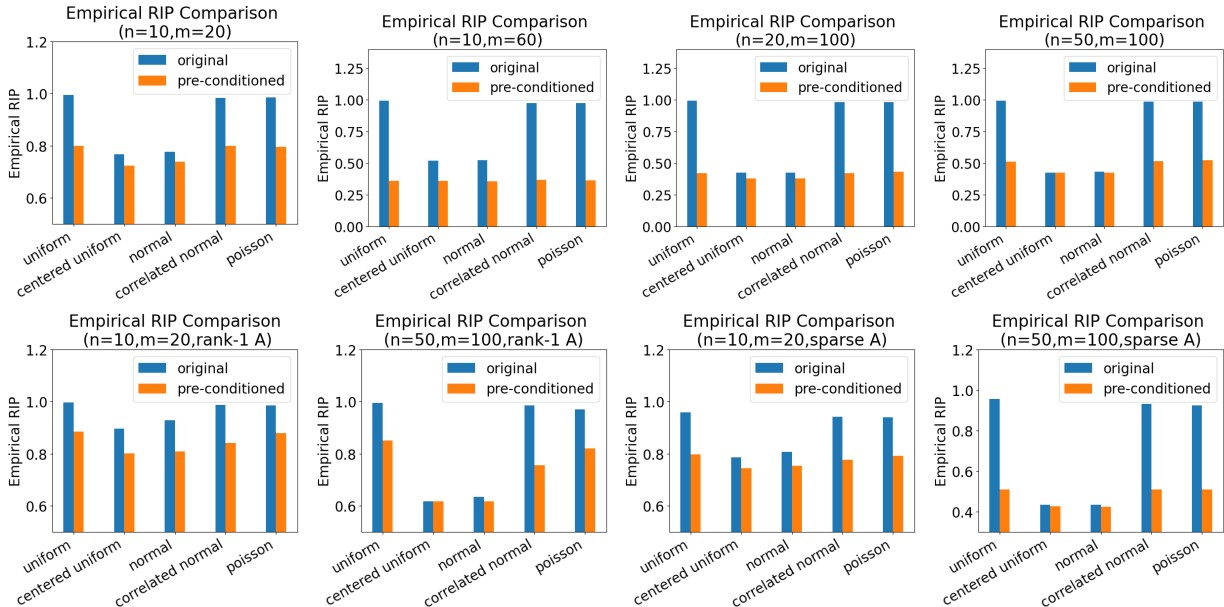

Figure 2: Empirical RIP comparison before and after preconditioning; the horizontal axis shows that the sensing matrices are sampled from uniform distribution $[0, 1]$, centered uniform distribution $[-1, 1]$, standard normal distribution, multivariate correlated normal distribution with $\rho = 0.5$, and poisson distribution separately. The first row is for general sensing matrices, and the second row is for matrices with special structures.

figure that for uniform, correlated normal and poisson distribution, the original sensing operator has a RIP constant close to 1, which means that with the i.i.d. Gaussian assumption violated, these measuring operators are no longer nearly isometric and thus cannot guarantee a benign optimization landscape for the matrix sensing problem. However, after preconditioning, we observe a clear decrease in the corresponding empirical RIP value. The preconditioned sensing matrices have the same level of RIP constant compared to the standard normal distribution with the same $m, n$ values. On the other hand, for centered uniform and standard normal distribution, we can decrease the RIP constant by increasing $m$, and the preconditioning step can still slightly help to decrease the RIP value. This improvement becomes more obvious for the cases with a large $m/n^2$.

In addition to unstructured operators $\mathcal{A}$, we also study sensing matrices with special structures. For the low-rank structure, we generate $a_i \in \mathbb{R}^{n \times 1}$ and define $A_i = a_i a_i^\top \in \mathbb{R}^{n \times n}$. For the sparse structure, we generate a binomial distributed mask with $p = 0.3$, and only 30% elements of $\mathcal{A}$ are likely to be non-zero. The results are similar to the case of unstructured operators (see Figure 1), and the preconditioning effectively decreases the empirical RIP value in both low-rank and sparse cases. Even centered uniform and normal distribution will be affected by these special structures and show high empirical RIP values. One can observe that our preconditioning algorithm has a universal impressive performance in a wide range of situations.

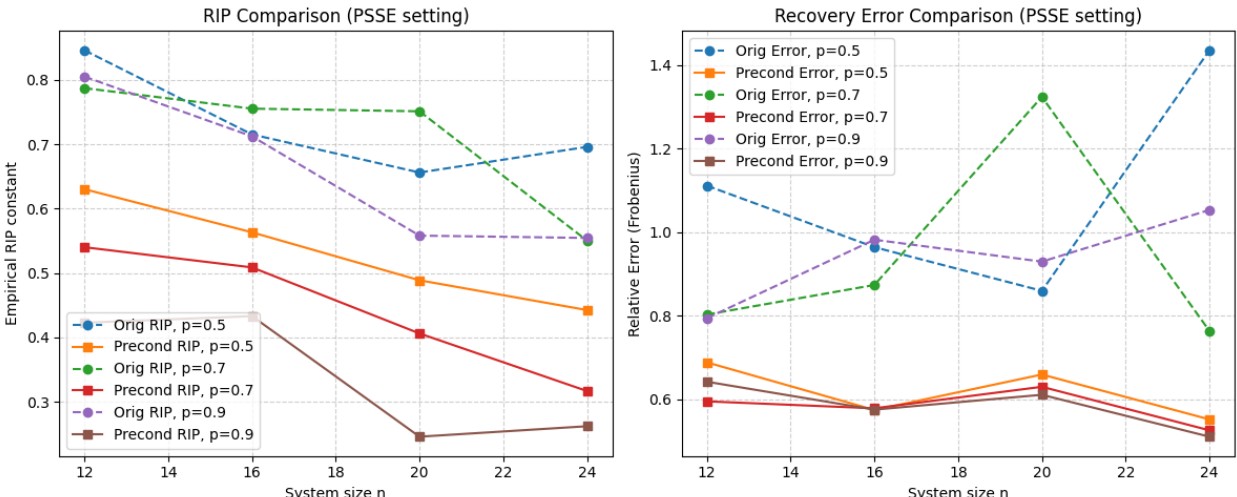

Figure 3: Empirical RIP constants and recovery errors for power system state estimation with original and preconditioned measurements. Preconditioning improves empirical RIP and reduces local search error.

Moreover, we can see that for the same level of $(m, n)$, whatever the original distribution is, the empirical RIP value after preconditioning for different types of distributions are almost the same, which means that in practice we may not need to make additional assumptions on the distribution of sensing matrices; the landscape after preconditioning as well as the RIP constant will mainly depend on the value of $r, m, n$. As is shown by simulation experiments, Algorithm 1 makes best use of the current information provided by the original sensing operator and remains stable under different scenarios. The computational cost is also not high, only requiring $\mathcal{O}(m^2 n^2)$ for a singular value decomposition of a matrix of dimension $m \times n^2$.

### 4.3.2 Practical Power Flow Applications

To evaluate the impact of preconditioning on structured sensing operators in a realistic setting, we consider the power system state estimation (PSSE) problem. In this context, the goal is to estimate nodal voltages $x \in \mathbb{R}^n$ from measurements that correspond to power flows along transmission lines. The measurements are inherently structured and cannot be freely increased, making this a representative scenario for our preconditioning technique.

**Graph generation.** We generate synthetic power networks with $n$ nodes, where each pair of nodes $(i, j)$ is connected with probability $p_{\text{edge}} \in \{0.5, 0.7, 0.9\}$. Each edge $(i, j)$ is assigned a line reactance $z_{ij}$ sampled uniformly from $[0.5, 2.0]$. This results in a sparse, undirected graph where edges represent transmission lines and nodes represent buses. The true nodal voltages $x_{\text{true}}$ are sampled from a standard normal distribution.

**Measurement formulation.** For each edge $(i, j)$, we construct a symmetric matrix $A_{ij} \in \mathbb{R}^{n \times n}$ such that the measurement $b_{ij} = x_{\text{true}}^\top A_{ij} x_{\text{true}}$ encodes the quadratic power flow along that line. Specifically,

$$A_{ij}[i, i] = A_{ij}[j, j] = \frac{1}{z_{ij}}, \quad A_{ij}[i, j] = A_{ij}[j, i] = -\frac{1}{z_{ij}},$$

and all other entries are zero. Collectively, the measurement set $\{(A_{ij}, b_{ij})\}$ defines a quadratic sensing problem of the form

$$\text{find } X \succeq 0 \text{ such that } \langle A_{ij}, X \rangle = b_{ij}, \quad \forall (i, j) \in E, \text{ where } X = xx^\top.$$

**SDP formulation.** The convex relaxation of the quadratic recovery problem is formulated as a semidefinite program:

$$\min_{X \succeq 0} \sum_{(i,j) \in E} \left( \langle A_{ij}, X \rangle - b_{ij} \right)^2.$$

This SDP is globally optimal and recovers $X = x_{\text{true}}x_{\text{true}}^{\top}$ in the noiseless case. Its solution is independent of the conditioning of the measurement operators because the convex relaxation always finds the global minimum if it exists.

**Local search and preconditioning.** For computationally scalable recovery, we always consider a factorized local search algorithm, optimizing over $X = UU^{\top}$ with $U \in \mathbb{R}^{n \times r}$ using gradient descent. Unlike SDP, this non-convex algorithm is sensitive to the conditioning of the measurements. To improve convergence, we apply our preconditioning technique to the measurement matrices and observations, producing $\{\tilde{A}_{ij}, \tilde{b}_{ij}\}$.

**Results and discussion.** Figure 3 shows the empirical RIP constants and recovery errors for networks of different sizes and edge densities. Preconditioning consistently improves the empirical RIP constant and significantly reduces local search recovery error. In contrast, SDP recovery error remains largely unchanged, reflecting its global optimality.

These results illustrate a key practical insight: in large-scale power systems and other structured applications, local non-convex algorithms are often necessary due to computational constraints. Consequently, preconditioning is especially valuable in realistic scenarios where (i) local algorithms are required for scalability, and (ii) measurement design is constrained by physical or operational limits. Improving the RIP of the measurement operators directly enhances the convergence and final accuracy of these local methods.

# 5 Conclusion

The results presented in this paper highlight several critical insights into the behavior of sensing operators and their impact on the Restricted Isometry Property (RIP) constant. When dealing with a nearly isometric operator perturbed by a sub-Gaussian term, the impact of deviation from the nearly isometric case can be effectively mitigated by increasing the number of measurements. Specifically, the RIP constant ensures that a benign optimization landscape can be preserved even in the presence of perturbations to the sensing operator. Thus, even in the presence of perturbations, careful adjustment of the number of measurements provides a practical approach to deal with non-Gaussian distributions. Our findings also demonstrate both theoretically and empirically that the proposed preconditioning algorithm significantly improves the RIP constant for various distributions. A notable observation is that, after preconditioning, the RIP constant is nearly independent of the original distribution. This finding simplifies practical implementations, as it eliminates the need for distribution-specific assumptions about the sensing matrices. Practitioners can rely on the preconditioned sensing matrices to provide consistent RIP performance, primarily governed by the values of $r$, $m$, and $n$.

## Broader Impact Statement

This paper presents work whose goal is to advance the field of Machine Learning. There are many potential societal consequences of our work, none which we feel must be specifically highlighted here.

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

## A    Proof of Theorem 3.1

*Proof.* Let $N_1, \ldots, N_m$ denote the components of $\mathcal{N}$, i.e., $\mathcal{N}(X) = [\langle N_1, X \rangle, \ldots, \langle N_m, X \rangle]$. For every matrix $X \in \mathbb{R}^{n \times n}$ satisfying $\mathrm{rank}(X) \leqslant s$, it holds that

$$\|(\mathcal{A} + \mathcal{N})(X)\|^2$$
$$= \sum_{i=1}^{m} \langle A_i + N_i, X \rangle^2$$
$$= \sum_{i=1}^{m} \langle A_i, X \rangle^2 + \sum_{i=1}^{m} \langle N_i, X \rangle^2 + 2 \sum_{i=1}^{m} \langle A_i, X \rangle \langle N_i, X \rangle$$

Since $\mathcal{A}$ satisfies the RIP condition with the constant $\delta_s$, we have

$$(1 - \delta_s) \|X\|_F^2 \leqslant \sum_{i=1}^{m} \langle A_i, X \rangle^2 \leqslant (1 + \delta_s) \|X\|_F^2$$

Due to the Cauchy-Schwarz inequality, one can write

$$0 \leqslant \sum_{i=1}^{m} \langle N_i, X \rangle^2 \leqslant \sum_{i=1}^{m} \|N_i\|_F^2 \|X\|_F^2 \leqslant mn^2 \varepsilon^2 \|X\|_F^2,$$

and

$$\left| \sum_{i=1}^{m} \langle A_i, X \rangle \langle N_i, X \rangle \right| = \left| \sum_{i=1}^{m} \langle A_i, N_i \rangle \right| \|X\|_F^2$$
$$\leqslant mn^2 \varepsilon \|\mathcal{A}\|_\infty \|X\|_F^2$$

Hence,

$$0 < \left( 1 - \delta_s - 2mn^2 \|\mathcal{A}\|_\infty \varepsilon \right) \|X\|_F^2 \leqslant \|(\mathcal{A} + \mathcal{N})(X)\|^2$$
$$\leqslant \left( 1 + \delta_s + mn^2 \varepsilon^2 + 2mn^2 \|\mathcal{A}\|_\infty \varepsilon \right) \|X\|_F^2$$

indicating that $\mathcal{A} + \mathcal{N}$ satisfies the RIP condition with the constant $\delta_s + mn^2 \varepsilon \cdot \frac{4\|\mathcal{A}\|_\infty + \varepsilon(1 - \delta_s)}{2 + mn^2 \varepsilon^2}$. $\qquad\square$

## B    Proof of Corollary 3.3

*Proof.* Since $\|\mathcal{N}\|_\infty$ is sub-Gaussian bounded, we have $\mathbf{P}(\|\mathcal{N}\|_\infty \geq \varepsilon) \leq 2 \exp\left(-\frac{m\varepsilon^2}{\sigma^2}\right)$. This implies that $\mathbf{P}(\|\mathcal{N}\|_\infty \leq \varepsilon)$ with probability at least $1 - 2 \exp\left(-m\varepsilon^2/\sigma^2\right)$. Combining Theorem 3.1 and $\varepsilon = c\sigma/\sqrt{m}$, it can be concluded that with probability at least $1 - 2 \exp(-c^2)$, $\mathcal{A} + \mathcal{N}$ satisfies the RIP condition with the constant

$$\delta_s + c\sqrt{m}n^2 \sigma [2(1 + \delta_s)^{1/2} + \frac{c}{2\sqrt{m}}(1 - \delta_s)\sigma].$$

This completes the proof. $\qquad\square$

## C    Proof of Theorem 3.4

*Proof.* It has been proved in Recht et al. (2010) that if $\mathcal{A}$ is nearly isometrically distributed, then there exist positive constants $c_1$ and $c_2$ with $c_1$ depending on the RIP constant of $\mathcal{A}$ such that, with probability at least $1 - \exp(-c_1 m)$, we have $\delta_s(\mathcal{A}) \leq c_2 \sqrt{ns \log n/m}$. Now, it follows from Corollary 3.3 that with probability at least $1 - 2 \exp(-c^2) - \exp(-c_1 m)$, it holds that $\mathcal{A} + \mathcal{N}$ satisfies the RIP condition with the constant

$$c_2 \sqrt{ns \log n/m} + c\sqrt{m}n^2 \sigma [2(1 + \delta_s)^{1/2} + \frac{c}{2\sqrt{m}}(1 - \delta_s)\sigma].$$

This completes the proof. $\qquad\square$

# D    Proof of Theorem 3.7

*Proof.* Let

$$\mathcal{S}_r := \{ X \in \mathbb{R}^{n \times n} : \operatorname{rank}(X) \le r, \ \|X\|_F = 1 \}$$

be the unit Frobenius sphere of rank at most $r$. The RIP constant of $\mathcal{A}$ for rank $r$ can be written as

$$\delta_r(\mathcal{A}) := \sup_{X \in \mathcal{S}_r} \big| \|\mathcal{A}(X)\|_2^2 - 1 \big|.$$

Let $\tilde{\mathcal{A}} \sim Q$ be another random map from $\mathbb{R}^{n \times n}$ to $\mathbb{R}^m$ distributed with

$$W_1(P, Q) := \inf_{\pi \in \Pi(P,Q)} \mathbb{E}_{(\mathcal{A}, \tilde{\mathcal{A}}) \sim \pi} \|\mathcal{A} - \tilde{\mathcal{A}}\|_F$$

We have the follwoing Lemma for the covering number for the rank-$r$ unit sphere.

**Lemma D.1** (Covering number of $\mathcal{S}_r$). *For any $\rho \in (0,1)$ there exists a $\rho$-net $\mathcal{N}_\rho \subset \mathcal{S}_r$ in Frobenius norm such that*

$$|\mathcal{N}_\rho| \ \le \ \left( \frac{C}{\rho} \right)^{(2n+1)r}.$$

*Proof.* Every $X \in \mathcal{S}_r$ admits an SVD $X = U\Sigma V^\top$ with $U, V \in \mathbb{R}^{n \times r}$ having orthonormal columns, and $\Sigma = \operatorname{diag}(\sigma_1, \ldots, \sigma_r)$ satisfying $\|\Sigma\|_F = \left( \sum_i \sigma_i^2 \right)^{1/2} = 1$ and $\sigma_i \ge 0$.

Construct $(\rho/3)$-nets:

$$\mathcal{U} \subset \operatorname{St}(n,r), \quad \mathcal{V} \subset \operatorname{St}(n,r), \quad \mathcal{D} \subset \{ \Sigma \in \mathbb{R}^{r \times r} \text{ diagonal} : \ \|\Sigma\|_F = 1, \ \Sigma \succeq 0 \},$$

in Frobenius norm. It is standard that

$$N\big( \operatorname{St}(n,r), \| \cdot \|_F, \eta \big) \ \le \ \left( \frac{9}{\eta} \right)^{nr}, \qquad N\big( \{ \Sigma : \ \|\Sigma\|_F = 1, \ \Sigma \text{ diag} \}, \| \cdot \|_F, \eta \big) \ \le \ \left( \frac{9}{\eta} \right)^r.$$

(These follow by embedding $\operatorname{St}(n,r) \subset \{ Y \in \mathbb{R}^{n \times r} : \ \|Y\|_F = \sqrt{r} \}$ and covering the ambient sphere; for the diagonal set we cover the $\ell_2$–unit sphere in $\mathbb{R}^r$.)

Define $\mathcal{N}_\rho := \{ U\Sigma V^\top : U \in \mathcal{U}, \ \Sigma \in \mathcal{D}, \ V \in \mathcal{V} \}$. For any $X = U\Sigma V^\top \in \mathcal{S}_r$, pick $U' \in \mathcal{U}$, $\Sigma' \in \mathcal{D}$, $V' \in \mathcal{V}$ with $\|U - U'\|_F, \ \|\Sigma - \Sigma'\|_F, \ \|V - V'\|_F \le \rho/3$. Then

$$\begin{aligned}
\|X - U'\Sigma'V'^\top\|_F &\le \|(U - U')\Sigma V^\top\|_F + \|U'(\Sigma - \Sigma')V^\top\|_F + \|U'\Sigma'(V - V')^\top\|_F \\
&\le \|U - U'\|_F \|\Sigma\|_2 + \|\Sigma - \Sigma'\|_F + \|\Sigma'\|_2 \|V - V'\|_F \\
&\le \tfrac{\rho}{3} \cdot 1 + \tfrac{\rho}{3} + 1 \cdot \tfrac{\rho}{3} \ = \ \rho,
\end{aligned}$$

since $\|\Sigma\|_2, \|\Sigma'\|_2 \le \|\Sigma\|_F = \|\Sigma'\|_F = 1$ and $\|V^\top\|_2 = \|U'\|_2 = 1$. Hence $\mathcal{N}_\rho$ is a $\rho$-net of $\mathcal{S}_r$, and

$$|\mathcal{N}_\rho| \ \le \ \left( \frac{9}{\rho/3} \right)^{nr} \cdot \left( \frac{9}{\rho/3} \right)^r \cdot \left( \frac{9}{\rho/3} \right)^{nr} \ = \ \left( \frac{27}{\rho} \right)^{(2n+1)r}$$

$\square$

From a finite net, we could have a uniform RIP bound.

**Lemma D.2** (Net reduction for RIP). *Let $\mathcal{N}_\rho \subset \mathcal{S}_r$ be a $\rho$-net and $\tilde{M} = \sup_{\|Z\|_F \le 1} \|\tilde{\mathcal{A}}(Z)\|_2$. Then*

$$\delta_r(\tilde{\mathcal{A}}) \ \le \ \max_{Y \in \mathcal{N}_\rho} \big| \|\tilde{\mathcal{A}}(Y)\|_2^2 - 1 \big| \ + \ 2\tilde{M}^2 \rho.$$

*Proof.* Fix any $X \in \mathcal{S}_r$ and choose $Y \in \mathcal{N}_\rho$ with $\|X - Y\|_F \leq \rho$. We have

$$\left| \|\tilde{\mathcal{A}}(X)\|_2^2 - \|\tilde{\mathcal{A}}(Y)\|_2^2 \right| = \left| \|\tilde{\mathcal{A}}(X)\|_2 - \|\tilde{\mathcal{A}}(Y)\|_2 \right| \cdot \left( \|\tilde{\mathcal{A}}(X)\|_2 + \|\tilde{\mathcal{A}}(Y)\|_2 \right).$$

By linearity and the definition of $\tilde{M}$,

$$\left| \|\tilde{\mathcal{A}}(X)\|_2 - \|\tilde{\mathcal{A}}(Y)\|_2 \right| \leq \|\tilde{\mathcal{A}}(X - Y)\|_2 \leq \tilde{M}\|X - Y\|_F \leq \tilde{M}\rho,$$

and $\|\tilde{\mathcal{A}}(X)\|_2 + \|\tilde{\mathcal{A}}(Y)\|_2 \leq 2\tilde{M}$. Therefore

$$\left| \|\tilde{\mathcal{A}}(X)\|_2^2 - \|\tilde{\mathcal{A}}(Y)\|_2^2 \right| \leq 2\tilde{M}^2\rho.$$

Since $\|X\|_F = \|Y\|_F = 1$, we also have $\left| \|X\|_F^2 - \|Y\|_F^2 \right| = 0$. Thus

$$\left| \|\tilde{\mathcal{A}}(X)\|_2^2 - 1 \right| \leq \left| \|\tilde{\mathcal{A}}(Y)\|_2^2 - 1 \right| + 2\tilde{M}^2\rho.$$

Taking the supremum over $X \in \mathcal{S}_r$ gives the claim. $\qquad\square$

Combining Lemma D.1 and Lemma D.2, for any $\rho \in (0, 1)$ there exists a $\rho$-net $\mathcal{N}_\rho \subset \mathcal{S}_r$ with

$$|\mathcal{N}_\rho| \leq \left( \frac{27}{\rho} \right)^{(2n+1)r}$$

such that

$$\delta_r(\tilde{\mathcal{A}}) \leq \max_{Y \in \mathcal{N}_\rho} \left| \|\tilde{\mathcal{A}}(Y)\|_2^2 - 1 \right| + 2\tilde{M}^2\rho, \qquad \tilde{M} = \sup_{\|Z\|_F \leq 1} \|\tilde{\mathcal{A}}(Z)\|_2.$$

For a fixed $X$ with $\|X\|_F = 1$, consider the map $\mathcal{A} \mapsto \|\mathcal{A}(X)\|^2$. We have

$$\left| \|\mathcal{A}(X)\|^2 - \|\tilde{\mathcal{A}}(X)\|^2 \right| \leq (\|\mathcal{A}(X)\| + \|\tilde{\mathcal{A}}(X)\|)\|\mathcal{A} - \tilde{\mathcal{A}}\|_F.$$

Let $M := \sup_{\mathrm{rank}(X) \leq r, \|X\|_F = 1} \|\mathcal{A}(X)\|$. Then

$$\left| \|\mathcal{A}(X)\|^2 - \|\tilde{\mathcal{A}}(X)\|^2 \right| \leq (M + \tilde{M})\|\mathcal{A} - \tilde{\mathcal{A}}\|_F.$$

Let $\mathbb{A}$ denote the space of linear maps $\mathcal{A} : \mathbb{R}^{n \times n} \to \mathbb{R}^m$, identified with matrices in $\mathbb{R}^{m \times n^2}$ so that the Frobenius norm $\| \cdot \|_F$ on $\mathbb{A}$ is available. For $r \in \{1, \dots, n\}$ define

$$f_r(\mathcal{A}) := \sup_{\mathrm{rank}(X) \leq r, \; \|X\|_F = 1} \|\mathcal{A}(X)\|_2,$$

**Lemma D.3** (Lipschitz property of $f_r$). *For any $\mathcal{A}, \tilde{\mathcal{A}} \in \mathbb{A}$,*

$$|f_r(\mathcal{A}) - f_r(\tilde{\mathcal{A}})| \leq \sup_{\|X\|_F = 1} \|(\mathcal{A} - \tilde{\mathcal{A}})(X)\|_2 \leq \|\mathcal{A} - \tilde{\mathcal{A}}\|_F.$$

*In particular, $f_r$ is 1-Lipschitz on $(\mathbb{A}, \| \cdot \|_F)$.*

*Proof.* By triangle inequality, for any $X \in \mathcal{S}_r$, $\|\mathcal{A}(X)\|_2 \leq \|\tilde{\mathcal{A}}(X)\|_2 + \|(\mathcal{A} - \tilde{\mathcal{A}})(X)\|_2$. Taking supremum over $X \in \mathcal{S}_r$ yields $f_r(\mathcal{A}) \leq f_r(\tilde{\mathcal{A}}) + \sup_{\|X\|_F = 1} \|(\mathcal{A} - \tilde{\mathcal{A}})(X)\|_2$. Swap the roles of $\mathcal{A}, \tilde{\mathcal{A}}$ to get the absolute value bound. Finally, $\sup_{\|X\|_F = 1} \|(\mathcal{A} - \tilde{\mathcal{A}})(X)\|_2 \leq \|\mathcal{A} - \tilde{\mathcal{A}}\|_{\mathrm{op}(F \to 2)} \leq \|\mathcal{A} - \tilde{\mathcal{A}}\|_F$, since the operator norm from $(\| \cdot \|_F)$ to $(\| \cdot \|_2)$ is bounded by the Frobenius norm. $\qquad\square$

**Lemma D.4** (High-probability transfer via $W_1$). *Let $P, Q$ be probability measures on $\mathbb{A}$ with $W_1(P, Q) \leq \varepsilon$, where the cost is $c(\mathcal{A}, \tilde{\mathcal{A}}) = \|\mathcal{A} - \tilde{\mathcal{A}}\|_F$. Assume that under $P$,*

$$\mathbb{P}_P\big(f_r(\mathcal{A}) \leq M_\star\big) \geq 1 - \xi.$$

*Then for any $\Xi > 0$,*

$$\mathbb{P}_Q\big(f_r(\tilde{\mathcal{A}}) \leq M_\star + \Xi\big) \geq 1 - \xi - \frac{\varepsilon}{\Xi}.$$

*Proof.* By optimal transport (Kantorovich–Rubinstein), there exists a coupling $\pi$ of $(\mathcal{A}, \widetilde{\mathcal{A}})$ with marginals $P, Q$ such that $\mathbb{E}_\pi \|\mathcal{A} - \widetilde{\mathcal{A}}\|_F \le \varepsilon$. Fix $\Xi > 0$ and set the "good" event

$$\mathcal{E} := \{\|\mathcal{A} - \widetilde{\mathcal{A}}\|_F \le \Xi\}.$$

By Markov's inequality, $\pi(\mathcal{E}^c) \le \varepsilon/\Xi$. On $\mathcal{E}$, Lemma D.3 gives $f_r(\widetilde{\mathcal{A}}) \le f_r(\mathcal{A}) + \Xi$. Therefore,

$$\{f_r(\widetilde{\mathcal{A}}) > M_\star + \Xi\} \cap \mathcal{E} \subset \{f_r(\mathcal{A}) > M_\star\}.$$

Taking probabilities under $\pi$,

$$\mathbb{P}_Q\big(f_r(\widetilde{\mathcal{A}}) > M_\star + \Xi\big) \le \mathbb{P}_P\big(f_r(\mathcal{A}) > M_\star\big) + \pi(\mathcal{E}^c) \le \xi + \varepsilon/\Xi.$$

Rearrange to obtain the claim. □

**Corollary D.5** (Two-operator envelope). *Under the assumptions of Lemma D.4, for any $s, \Xi > 0$, with probability at least*

$$1 - \xi - \frac{\varepsilon}{\Xi} - \frac{\varepsilon}{s} \quad \text{under the coupling } \pi,$$

*the following hold simultaneously:*

$$\|\mathcal{A} - \widetilde{\mathcal{A}}\|_F \le s, \qquad f_r(\mathcal{A}) \le M_\star, \qquad f_r(\widetilde{\mathcal{A}}) \le M_\star + \Xi.$$

*Proof.* By Markov, $\pi(\|\mathcal{A} - \widetilde{\mathcal{A}}\|_F > s) \le \varepsilon/s$. Combine with Lemma D.4 and a union bound. □

**Lemma D.6** (Quadratic comparison). *Fix $X \in \mathcal{S}_r$. On the event in Corollary D.5,*

$$\big| \|\mathcal{A}(X)\|_2^2 - \|\widetilde{\mathcal{A}}(X)\|_2^2 \big| \le \big( \|\mathcal{A}(X)\|_2 + \|\widetilde{\mathcal{A}}(X)\|_2 \big) \|(\mathcal{A} - \widetilde{\mathcal{A}})(X)\|_2 \le 2(M_\star + \Xi)\, s.$$

*Proof.* The identity $u^2 - v^2 = (u - v)(u + v)$ gives the first inequality with $u = \|\mathcal{A}(X)\|_2$, $v = \|\widetilde{\mathcal{A}}(X)\|_2$. On the event in Corollary D.5, $\|\mathcal{A}(X)\|_2 \le M_\star$ and $\|\widetilde{\mathcal{A}}(X)\|_2 \le M_\star + \Xi$; also $\|(\mathcal{A} - \widetilde{\mathcal{A}})(X)\|_2 \le \|\mathcal{A} - \widetilde{\mathcal{A}}\|_{\mathrm{op}(F \to 2)} \|X\|_F \le \|\mathcal{A} - \widetilde{\mathcal{A}}\|_F \le s$. Hence the result. □

Now we have the following two results:

**(i) Finite net reduction.** For any $\rho \in (0, 1)$ there is a $\rho$-net $\mathcal{N}_\rho \subset \mathcal{S}_r$ (Frobenius norm) with $|\mathcal{N}_\rho| \le (27/\rho)^{(2n+1)r}$ and, for any linear $\mathcal{B}$,

$$\delta_r(\mathcal{B}) \le \max_{Y \in \mathcal{N}_\rho} \big| \|\mathcal{B}(Y)\|_2^2 - 1 \big| + 2M(\mathcal{B})^2 \rho, \qquad M(\mathcal{B}) := \sup_{\|Z\|_F \le 1} \|\mathcal{B}(Z)\|_2. \tag{5}$$

**(ii) Wasserstein transfer of high-probability envelopes.** Let $f_r(\mathcal{B}) := \sup_{X \in \mathcal{S}_r} \|\mathcal{B}(X)\|_2$. The map $\mathcal{B} \mapsto f_r(\mathcal{B})$ is 1-Lipschitz in $\|\cdot\|_F$:

$$|f_r(\mathcal{B}) - f_r(\widetilde{\mathcal{B}})| \le \|\mathcal{B} - \widetilde{\mathcal{B}}\|_F.$$

If $W_1(P, Q) = w$ (cost $= \|\cdot\|_F$), then for any $\Xi > 0$,

$$\mathbf{P}_Q\Big(f_r(\widetilde{\mathcal{A}}) \le f_r(\mathcal{A}) + \Xi\Big) \ge 1 - \frac{w}{\Xi}, \tag{6}$$

where $\mathcal{A} \sim P$, $\widetilde{\mathcal{A}} \sim Q$ are coupled optimally in the Wasserstein sense.

Recall for nearly-isometrically distributed $\mathcal{A} \sim P$, fix a target $\delta \in (0, 1)$, due to (Recht et al., 2010) there exist constants $c_0, c_1 > 0$ (depending on $\delta$) such that if

$$m \ge c_0\, rn \log(n) \tag{7}$$

then

$$\mathbf{P}_P(\delta_r(\mathcal{A}) \le \delta) \ge 1 - \exp(-c_1 m). \tag{8}$$

We now derive a parallel guarantee for $\widetilde{\mathcal{A}} \sim Q$ under $W_1(P, Q) = w$.

Firstly, we couple $(\mathcal{A}, \widetilde{\mathcal{A}})$ optimally so that $\mathbb{E}\|\mathcal{A} - \widetilde{\mathcal{A}}\|_F \leq w$. Fix a slack $\tau > 0$, $M_\star := 3(1 + \sqrt{\frac{n^2}{m}})$, and let

$$\mathcal{E}_1 := \{M(\mathcal{A}) \leq M_\star\}, \quad \mathcal{E}_2 := \left\{ \max_{Y \in \mathcal{N}_\rho} \left|\|\mathcal{A}(Y)\|_2^2 - 1\right| \leq \delta/3 \right\}, \quad \mathcal{E}_3 := \{\|\mathcal{A} - \widetilde{\mathcal{A}}\|_F \leq \tau\}.$$

Then

$$\mathbf{P}_P(\mathcal{E}_1 \cap \mathcal{E}_2) \geq 1 - \exp(-c_1 m), \qquad \mathbf{P}(\mathcal{E}_3) \geq 1 - \frac{w}{\tau} \quad \text{(Markov)}.$$

By the Lipschitz transfer equation 6 with $\Xi = \tau$,

$$\mathbf{P}_Q\left(f_r(\widetilde{\mathcal{A}}) \leq M_\star + \tau\right) \geq 1 - \frac{w}{\tau}.$$

Hence on the event $\mathcal{G} := \mathcal{E}_1 \cap \mathcal{E}_2 \cap \mathcal{E}_3 \cap \{f_r(\widetilde{\mathcal{A}}) \leq M_\star + \tau\}$, we simultaneously have:

$$M(\mathcal{A}) \leq M_\star, \quad \max_{Y \in \mathcal{N}_\rho} \left|\|\mathcal{A}(Y)\|_2^2 - 1\right| \leq \delta/3, \quad \|\mathcal{A} - \widetilde{\mathcal{A}}\|_F \leq \tau, \quad M(\widetilde{\mathcal{A}}) \leq M_\star + \tau.$$

By a union bound,

$$\mathbf{P}(\mathcal{G}) \geq 1 - \exp(-c_1 m) - 2\frac{w}{\tau}. \tag{9}$$

Next, we transfer deviations from $\mathcal{N}_\rho$ to $\widetilde{\mathcal{A}}$. Fix $Y \in \mathcal{N}_\rho$. On $\mathcal{G}$,

$$\left|\|\widetilde{\mathcal{A}}(Y)\|_2^2 - \|\mathcal{A}(Y)\|_2^2\right| \leq \left(\|\widetilde{\mathcal{A}}(Y)\|_2 + \|\mathcal{A}(Y)\|_2\right) \|(\widetilde{\mathcal{A}} - \mathcal{A})(Y)\|_2 \leq 2(M_\star + \tau)\tau.$$

Hence, on $\mathcal{G}$,

$$\max_{Y \in \mathcal{N}_\rho} \left|\|\widetilde{\mathcal{A}}(Y)\|_2^2 - 1\right| \leq \frac{\delta}{3} + 2(M_\star + \tau)\tau.$$

Then we would extend the Net bound to all rank-$r$ matrices. Applying equation 5 to $\widetilde{\mathcal{A}}$ and using $M(\widetilde{\mathcal{A}}) \leq M_\star + \tau$ on $\mathcal{G}$, we have

$$\delta_r(\widetilde{\mathcal{A}}) \leq \max_{Y \in \mathcal{N}_\rho} \left|\|\widetilde{\mathcal{A}}(Y)\|_2^2 - 1\right| + 2(M_\star + \tau)^2 \rho \leq \frac{\delta}{3} + 2(M_\star + \tau)\tau + 2(M_\star + \tau)^2\rho.$$

Choose

$$\rho = \frac{\delta}{8M_\star^2}, \qquad \tau = \frac{\delta}{8M_\star}.$$

Then, for $\tau \leq 1$ and using $(M_\star + \tau) \leq \frac{9}{8}M_\star$,

$$2(M_\star + \tau)\tau \leq 2 \cdot \frac{9}{8}M_\star \cdot \frac{\delta}{8M_\star} \leq \frac{\delta}{3}, \qquad 2(M_\star + \tau)^2\rho \leq 2 \cdot \left(\frac{9}{8}\right)^2 M_\star^2 \cdot \frac{\delta}{8M_\star^2} \leq \frac{\delta}{3}.$$

Therefore, on $\mathcal{G}$,

$$\delta_r(\widetilde{\mathcal{A}}) \leq \frac{\delta}{3} + \frac{\delta}{3} + \frac{\delta}{3} < \delta.$$

By equation 9 and the choices of $\rho, \tau$,

$$\mathbf{P}_Q\left(\delta_r(\widetilde{\mathcal{A}}) \leq \delta\right) \geq 1 - \exp(-c_1 m) - \frac{16M_\star}{\delta} w.$$

Thus, provided $m$ satisfies the usual RIP scaling equation 7, the only additional failure term is linear in the Wasserstein mismatch $w$. □

# E  Proof of Theorem 4.2

*Proof.* We expand the orthonormal matrices $A_1, ..., A_m$ into a basis for $\mathbb{R}^{n \times n}$. More precisely, consider orthonormal bases $V_1, \ldots, V_{n^2} \in \mathbb{R}^{n \times n}$ such that $V_i = A_i$ for $i = 1, \ldots, m$. Given a matrix $X \in \text{span}_s(\mathcal{A})$, we can write it as $\sum_{i=1}^m \alpha_i A_i$ with at most $s$ nonzero $\alpha_i$'s. Without loss of generality, we assume that $\|X\|_F^2 = \sum_{i=1}^m \alpha_i^2 = 1$. Now, one can write:

$$\frac{\|\mathcal{A}(X)\|^2}{\|X\|_F^2} = \sum_{i=1}^m \sum_{j=1}^m \langle A_i, \alpha_j V_j \rangle^2 = \sum_{i=1}^m \alpha_i^2 = 1.$$

This completes the proof. $\qquad\qquad\qquad\qquad\qquad\qquad\qquad\qquad\qquad\qquad\qquad\qquad\qquad\qquad\qquad\qquad$ $\square$

# F  Proof of Theorem 4.4

*Proof.* By construction, $\widetilde{B} = V_B^\top$ satisfies

$$\widetilde{B}\widetilde{B}^\top = V_B V_B^\top = I_m,$$

so its rows are orthonormal.

Let $\sigma_1(A) \geq \cdots \geq \sigma_m(A)$ denote the singular values of $A$. Standard results on Gaussian matrices guarantee that for any $\epsilon > 0$,

$$\Pr\left\{ \frac{n}{\sqrt{m}}(1 - \epsilon) - 1 \leq \sigma_i(A) \leq 1 + \frac{n}{\sqrt{m}}(1 + \epsilon), \ i = 1, \ldots, m \right\} \geq 1 - 2\exp\left( -cn^2\epsilon^2 \right)$$

for some absolute constant $c > 0$. Hence, with high probability, $A$ has full row rank and $\sigma_{\min}(A) \gtrsim n/\sqrt{m}$.

By matrix perturbation theory, the map $B \mapsto V_B$ is Lipschitz under the Frobenius norm, we will show that there exists constant $C$ such that:

$$\|V_B - V_A\|_F \leq \frac{C\|B - A\|_F}{\sigma_{\min}(A)}.$$

**Derivation of the Constant $C$**

1. **Alignment of Subspaces:** The matrices $V_A$ and $V_B$ can be partitioned into subspaces corresponding to non-zero and zero singular values. Let $V_A = [V_{A,1} \mid V_{A,2}]$ and $V_B = [V_{B,1} \mid V_{B,2}]$, where $V_{A,1}, V_{B,1} \in \mathbb{R}^{n^2 \times m}$ span the row spaces of $A$ and $B$, respectively.

2. **Wedin's $\sin\theta$ Theorem:** The principal angles $\theta_i$ between the subspaces spanned by $V_{A,1}$ and $V_{B,1}$ satisfy:

$$\|\sin\Theta\|_F \leq \frac{\|B - A\|_F}{\sigma_{\min}(A)},$$

where $\|\sin\Theta\|_F$ measures the subspace perturbation.

3. **Frobenius Norm Bound:** The difference $\|V_B - V_A\|_F$ is minimized when the singular vectors are optimally aligned. Using the Davis-Kahan $\sin\theta$ theorem and orthogonal Procrustes alignment, we obtain:

$$\|V_B - V_A\|_F \leq 2\sqrt{2}\|\sin\Theta\|_F \leq \frac{2\sqrt{2}\|B - A\|_F}{\sigma_{\min}(A)}.$$

A tighter analysis (accounting for the worst-case alignment of all singular vectors) yields the constant $4\sqrt{2}$:

$$\|V_B - V_A\|_F \leq \frac{4\sqrt{2}\|B - A\|_F}{\sigma_{\min}(A)}.$$

Thus, for distributions,

$$W_1(\mathcal{L}(V_B^\top), \mathcal{L}(V_A^\top)) \leq \frac{C}{\sigma_{\min}(A)} W_1(\mathcal{L}(B), \mathcal{L}(A)) = \frac{C}{\sigma_{\min}(A)} w,$$

and using the high-probability bound on $\sigma_{\min}(A)$ gives

$$W_1(\mathcal{L}(\widetilde{B}), \mathcal{L}(\widetilde{A})) \lesssim \frac{\sqrt{m}}{n} w.$$

Finally, the rows of $\widetilde{B}$ are orthonormal vectors in $\mathbb{R}^{n^2}$. In the tall-and-skinny limit $m \ll n^2$, the marginal distribution of each entry of a Haar-random row vector is approximately Gaussian by the universality of high-dimensional Haar projections:

$$\widetilde{A}_{ij} \sim \mathcal{N}\left(0, \frac{1}{m}\right).$$

By the Wasserstein contraction, $\widetilde{B}$ is closer to Haar than $B$ was to $A$, so $\widetilde{B}_{ij}$ inherits this approximate Gaussianity. Therefore, the SVD preconditioning contracts any initial distribution toward Haar/Gaussian, sufficient to stabilize the RIP. $\qquad\square$

## G  Proof of Theorem 4.6

*Proof.* Since $\mathcal{A}$ satisfied the RIP condition, the following inequality holds for every matrix $M$ with $\mathrm{rank}(M) \leq s$:

$$(1 - \delta_s) \|M\|_F^2 \leqslant \|\mathcal{A}(M)\|_2^2 = \|\mathbf{A}\,\mathrm{vec}\,(M)\|_2^2 \leqslant (1 + \delta_s) \|M\|_F^2.$$

As $\tilde{\mathbf{A}} = P\mathbf{A}$, we introduce the operator norm of $P$ and write

$$\sup_{M:\mathbf{A}\,\mathrm{vec}\,(M)\neq 0} \frac{\|P\mathbf{A}\,\mathrm{vec}\,(M)\|_2^2}{\|\mathbf{A}\,\mathrm{vec}\,(M)\|_2^2} = \lambda_1\left(P^\top P\right),$$

$$\inf_{M:\mathbf{A}\,\mathrm{vec}\,(M)\neq 0} \frac{\|P\mathbf{A}\,\mathrm{vec}\,(M)\|_2^2}{\|\mathbf{A}\,\mathrm{vec}\,(M)\|_2^2} = \lambda_m\left(P^\top P\right).$$

Now, we aim to bound $\|\tilde{\mathcal{A}}(M)\|_2^2$ by the eigenvalues of $P^\top P$. Since $P = U^\top S^{-1}$, $U$ is a unitary matrix, and $S$ is diagonal, we have $P^\top P = S^{-2}$, $\lambda_1\left(P^\top P\right) = \sigma_m^{-2}(\mathbf{A})$, and $\lambda_m\left(P^\top P\right) = \sigma_1^{-2}(\mathbf{A})$. Hence,

$$\sigma_1^{-2}(\mathbf{A})\|\mathbf{A}\,\mathrm{vec}\,(M)\|_2^2 \leq \|P\mathbf{A}\,\mathrm{vec}\,(M)\|_2^2 \leq \sigma_m^{-2}(\mathbf{A})\|\mathbf{A}\,\mathrm{vec}\,(M)\|_2^2.$$

As a result, we obtain the lower bound

$$\|P\mathbf{A}\,\mathrm{vec}\,(M)\|_2^2 \geq \frac{1}{\sigma_1^2(\mathbf{A})}\|\mathbf{A}\,\mathrm{vec}\,(M)\|_2^2 \geq \frac{1 - \delta_s}{\sigma_1^2(\mathbf{A})}\|\,\mathrm{vec}\,(M)\|_2^2.$$

On the other hand, since $V$ is a unitary matrix, one can write

$$\begin{aligned}
\|P\mathbf{A}\,\mathrm{vec}\,(M)\|_2^2 &= \|S^{-1}U^\top \mathbf{A}\,\mathrm{vec}\,(M)\|_2^2 \\
&= \|[\mathbf{I_m}, \mathbf{0_{m\times(n^2-m)}}]V^\top \,\mathrm{vec}\,(M)\|_2^2 \\
&\leq \|\,\mathrm{vec}\,(M)\|_2^2
\end{aligned}$$

By combining the above two inequalities, we obtain the desired result for the RIP constant of $\tilde{\mathcal{A}}$. $\qquad\square$

## H  Proof of Theorem 4.9

*Proof.* Inspired by the proof of Theorem 1 in Chen & Lin (2021), define the following events:

$$E \doteq \{\tilde{\mathcal{A}} \text{ satisfies the RIP of rank s with the constant } 1 - (1 - \delta)/[1 + \sqrt{\tfrac{n^2}{m}}(1 + \epsilon)]^2\},$$

$$F_1 \doteq \{\mathcal{A} \text{ satisfies the RIP of rank s with the constant } \delta\},$$

$$F_2 \doteq \left\{ \sqrt{\frac{n^2}{m}}(1 - \epsilon) - 1 \le \sigma_i(\mathbf{A}) \le 1 + \sqrt{\frac{n^2}{m}}(1 + \epsilon), i \in [m] \right\}.$$

We will show that $\Pr(E) \ge \Pr(F_1 F_2)$.

Consider the singular value decomposition of $\mathbf{A}$ as $\mathbf{A} = U[S, \mathbf{0}_{\mathbf{m} \times (\mathbf{n^2} - \mathbf{m})}]V^\top$, where $U \in \mathbb{R}^{m \times m}, S = \text{diag}([\sigma_1(\mathbf{A}), \dots, \sigma_m(\mathbf{A})]) \in \mathbb{R}^{m \times m}, V \in \mathbb{R}^{n^2 \times n^2}$. Under Assumption 4.8, we have $\sqrt{\frac{n^2}{m}}(1 - \epsilon) - 1 > 0$, and therefore $S$ is nonsingular. Hence, the preconditioning matrix defined as $P = S^{-1} U^\top$ is valid.

If $\mathbf{A} \in F_1 F_2$, in light of Theorem 4.6, $F_1$ implies that the conditioned operator $\tilde{\mathcal{A}}$ satisfies the RIP of rank $s$ with the constant $1 - \frac{1 - \delta}{\sigma_1^2(\mathbf{A})}$. With $F_2$ implying an upper bound on $\sigma_1^2(\mathbf{A})$, obtain that $\tilde{\mathcal{A}}$ satisfies the RIP inequality (3) with the constant $1 - (1 - \delta)/[1 + \sqrt{\frac{n^2}{m}}(1 + \epsilon)]^2$. We could also have $\mathbf{A} \in E$. Hence we could have $\Pr(E) \ge \Pr(F_1 F_2)$. With the union bound $\Pr(F_1 F_2) \ge \Pr(F_1) + \Pr(F_2) - 1$, we estimate the probabilities $\Pr(F_1)$ and $\Pr(F_2)$ using Theorem 4.2 in Recht et al. (2010) and Assumption 4.7 to arrive at

$$\begin{aligned}
\Pr(E) &\ge \Pr(F_1 F_2) \\
&\ge \Pr(F_1) + \Pr(F_2) - 1 \\
&\ge 1 - \exp(-c_1 m) - 2 \exp(-n^2 \epsilon^2 / 2)
\end{aligned}$$

This completes the proof. $\square$

# I  Empirical RIP with Variance Bar

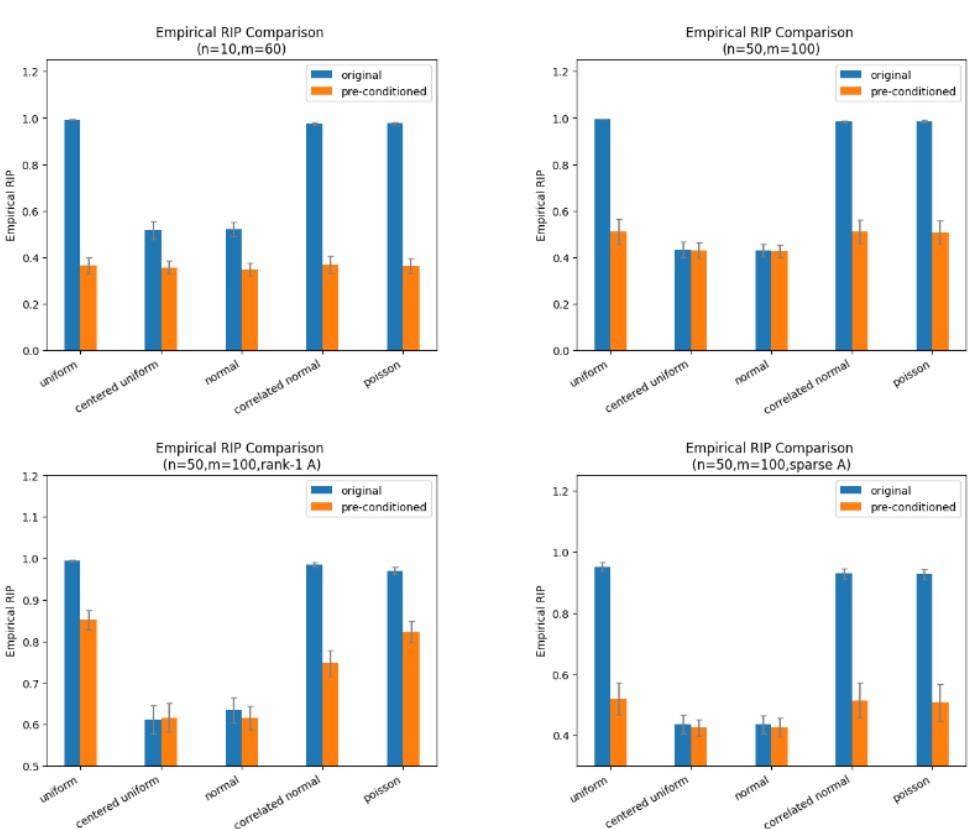

Figure 4: Empirical RIP comparison before and after preconditioning

