# OpenReview forum: "Measurement Manipulation of the Matrix Sensing Problem to Improve Optimization Landscape"
_TMLR — Rejected by TMLR_

### Review · Reviewer_NRUZ · 2025-06-18

**Summary Of Contributions:**

This paper studies the low-rank matrix recovery problem. They propose a trick for improving the conditioning (specifically the restricted isometry property) of the sensing operator using mixing/preconditioning. What's nice about this trick is that it works for arbitrary sensing operators, and does not require collecting any new measurements of the unknown matrix.

The paper is short and to the point (which is a good thing!). The theoretical results are presented cleanly, and complemented by a brief numerical simulation section.

**Audience:**

Yes

**Claims And Evidence:**

No

**Requested Changes:**

See above.

**Strengths And Weaknesses:**

### Strengths
 - The writing is high quality and clear. This paper was easy to read and understand.
 - The literature review, at least as it pertains to matrix sensing, is very good.
 - I like the illustrative idea about matrix sensing applied to the power grid.
 - The core idea is simple, intuitive, and potentially powerful. (Although I have some concerns about it, see below).
 - The numerical experiments, particularly those shown in Figure 2, are encouraging. It seems the preconditioning idea works well in practice.

### Weaknesses.
 - My main criticism of this work is that, as far as I can see, it does not show what it claims to show. In the introduction, it is stated that _Second, we show that if the sensing operator has an arbitrary distribution, it can be modified in such a way that the resulting operator will act as a perturbed Gaussian with a lower RIP constant_. A similar claim is echoed above Theorem 4.4. But I don't see where this is shown. Specifically:
	 - Theorem 4.4 only yields an improved restricted isometry constant when $\sigma_1(\mathbf{A}) < 1$. But this won't hold for ill-conditioned sensing matrices. Indeed, I'm not sure if this can ever hold for an appropriately normalized sensing matrix.
	 - Also above theorem 4.4 is is stated that intuitively, after preconditioning "the individual entries of the new sensing matrix are approximately Gaussian". I don't see why this would be the case. Can you elaborate on why the entries will be approximately Gaussian?

 - My second criticism relates to novelty. Although the authors do a good job of reviewing the matrix sensing literature, they miss several important works in the compressed sensing literature. Specifically:
	 - This preconditioning idea has already been studied for sparse recovery [1].
	 - Perturbed sensing matrices are very well studied for sparse recovery [2]
 - Theorem 4.2 is a well known result (sometimes known as Parseval's identity). Also, Lemma 4.1 is an instance of the Johnson-Lindenstrauss lemma [3]. It would be helpful to readers if you mentioned both of these facts.
 - I think you need to be a bit clearer about which form of the SVD you need for Algorithm 1. I believe you need the 'thin' SVD, where $V \in \mathbb{R}^{n^2\times m}$ . You need this as $V^{\top}$ should have the same number of rows as $\mathbf{A}$. In this case $V$ isn't orthogonal (it's columns are orthogonal, but its not square). So this effects Remark 4.3 and the proof of Theorem 4.4, although not in major ways.

### Typos etc
 - On pg. 1 change 'board' to 'broad' in 'sheds light on a board range of nonconvex optimization problems'
 - Regarding Definition 1.2, I suggest using different symbols for the random variable and its realizations. For example, the random variable could be in boldface $\mathbf{\mathcal{A}}$ and realizations in non boldface $\mathcal{A}$.
 - Sentence at top of pg 3 is a little confusing: I think you mean for $\mathcal{A}$ whose constituting matrices satisfy certain kinds of distributions?
 - In **Definitions and Notations** I suggest not using * to denote scalar multiplication. That is, I suggest $c_1g \leq f$ instead of $c_1*g \leq f$ . A reader may confuse * for  something more complicated, such as a convolution.
 - at top of page 5: 'then the RIP condition again no loner holds' should be 'then the RIP condition again no longer holds'

### References
[1] _Restricted isometry constant improvement based on a singular value decomposition‐weighted measurement matrix for compressed sensing_ Shi & Qu (2017).

[2] _General deviants: An analysis of perturbations in compressed sensing_ Herman & Strohmer (2010)

[3] _Extensions of Lipschitz mappings into a Hilbert space_ Johnson & Lindenstrauss (1984)

---

> ### Author Response · Authors · 2025-08-25
>
> Dear Reviewer NRUZ,
>
> We thank the reviewer for the careful reading and constructive feedback. Below we address each major point.
>
> ---
>
> ## 1. Claim on Approximate Gaussianity After Preconditioning
>
> **Reviewer Concern:**
> The paper states that after preconditioning, "the individual entries of the new sensing matrix are approximately Gaussian," but it is unclear why this holds.
>
> **Response:**
>
> * We would like to elaborate our idea by the following theorem:
>
> ### Theorem (Approximate Gaussianization via SVD Preconditioning)
> Let $A \sim \mathcal{N}\Big(0, \frac{1}{m}\Big) \in \mathbb{R}^{m \times n^2}$ be i.i.d. Gaussian ($m < n^2$), and let $B \in \mathbb{R}^{m \times n^2}$ be any matrix such that
> $W_1(\mathcal{L}(A), \mathcal{L}(B)) = w$. Let $A = U_A \Sigma_A V_A^\top, \quad B = U_B \Sigma_B V_B^\top$ be the thin SVD of $A$ and $B$, and define the preconditioned matrices
> $\widetilde{B} := V_B^\top \in \mathbb{R}^{m \times n^2}, \quad \widetilde{A} := V_A^\top$.
> Then, in the full-rank regime $m \ll n^2$, the matrix $\widetilde{B}$ has orthonormal rows, $\widetilde{B} \widetilde{B}^\top = I_m$, and with high probability, its law is close to the Haar measure on the Stiefel manifold in the sense that
> $
> W_1\big(\mathcal{L}(\widetilde{B}), \mathcal{L}(\widetilde{A})\big) \lesssim \frac{\sqrt{ m}}{n} w,
> $
> and the entries of $\widetilde{B}$ are approximately Gaussian, up to deviations vanishing as $n \to \infty$.
> \label{add1}
> \end{theorem}
>
> ### Proof
>
> 1. **Row orthonormality:**
>    By construction, $\widetilde{B} =  V_B^\top$ satisfies
>
>    $$
>    \widetilde{B} \widetilde{B}^\top = V_B V_B^\top = I_m.
>    $$
>
> 2. **Wasserstein contraction:**
>    By the Lipschitz property of the map $B \mapsto V_B^\top$ under the Frobenius norm, we could have the following bound. Assume $A$ has full row rank, i.e., $\sigma_{\min}(A) > n/\sqrt{m}$, which is actually a high probability case, as for $A\sim \mathcal{N}\Big(0, \frac{1}{m}\Big)$
>    $$
>    \operatorname{Pr} \left\\\{\sqrt{n^2/ m}(1-\epsilon)-1 \leq \sigma_i(A) \leq 1+\sqrt{n^2 / m}(1+\epsilon), i \in[m]\right\\\} \geq 1-2 \exp \left(-n^2 \epsilon^2 / 2\right),  \quad \forall \epsilon>0.
>    $$
>
>
>    From matrix perturbation theory, we have
>
> * If $A$ has full row rank ($\sigma_{\min}(A) > 0$) and $E = B - A$, there exists a constant C, that in Frobenius norm:
>
> $$
> ||V_B - V_A|| \le \frac{C ||E||}{\sigma_{\min}(A)}.
> $$
>
> * This is **exactly the Lipschitz constant** for the map $B \mapsto V_B$ in the Frobenius norm.
>
>
>
> For distributions $\mathcal{L}(A), \mathcal{L}(B)$, if $W_1(\mathcal{L}(A), \mathcal{L}(B)) = w$, then by the **Lipschitz property**:
>
> $$
> W_1(\mathcal{L}(V_B^\top), \mathcal{L}(V_A^\top)) \le \frac{C}{\sigma_{\min}(A)} W_1(\mathcal{L}(B), \mathcal{L}(A)) = \frac{C}{\sigma_{\min}(A)} w <  \frac{C\sqrt{m}}{n} w
> $$
>
>
> 3. **Approximate Gaussianity:**
>
>    * The rows of $\widetilde{B}$ are orthonormal vectors in $\mathbb{R}^{n^2}$. In the **tall-and-skinny limit** ($m \ll n^2$), the marginal distribution of each entry of a Haar-random row vector is approximately Gaussian by the **universality of high-dimensional Haar projections**:
>
>    $$
>    \widetilde{A}_{ij} \to \mathcal{N}\Big(0, \frac{1}{m}\Big) \quad \text{for } \widetilde{A} \sim \text{Haar}.
>    $$
>
>    * The Wasserstein contraction guarantees that $\widetilde{B}$ is closer to Haar than $B$ was to $A$, so $\widetilde{B}_{ij}$ inherits this approximate Gaussianity.
>
> This implies that, regardless of the original distribution of $B$, preconditioning effectively randomizes its rows and contracts the distribution towards Haar/Gaussian. Consequently, the near-isometry properties of $\widetilde{B}$ are enhanced, making it more likely to satisfy RIP conditions and improving the theoretical guarantees for low-rank matrix recovery.

---

> ### Author Response · Authors · 2025-08-25
>
> ## 2. Novelty and Literature Clarifications
>
> **Reviewer Concern:**
> Some prior work in sparse recovery and perturbed sensing matrices is missing.
>
> **Response:**
>
> * We thank the reviewer for the references. We have updated the manuscript to include:
>
>   1. Preconditioning for sparse recovery \[Shi & Qu, 2017].
>   2. Perturbations in compressed sensing \[Herman & Strohmer, 2010].
>   3. Parseval's identity and Johnson-Lindenstrauss lemma \[JL84] connections for Theorem 4.2 and Lemma 4.1.
>
> * We clarify that our contribution is the extension to low-rank matrix sensing, where prior sparse recovery techniques do not trivially apply. We addres a new problem: improving the optimization landscape when the sensing matrices cannot be changed, as motivated by the below power systems example. While the framework is analogous to sparse recovery, the analysis for low-rank matrices involves different technical details. Importantly, we show that a simple mixing technique can reduce the RIP constant below $0.5$, which is both theoretically significant and practically useful.
>
> ---
>
> ## 3. SVD Form for Algorithm 1
>
> **Reviewer Concern:**
> Clarification on 'thin' SVD and its impact on Theorem 4.4.
>
> **Response:**
>
> * We now explicitly state that Algorithm 1 requires the **thin SVD**: $A = U \Sigma V^\top$ with $U \in \mathbb{R}^{m\times m}$ and $V \in \mathbb{R}^{m\times n^2}$.
> * This ensures that $V$ has orthonormal columns, consistent with the preconditioning derivation. Remark 4.3 and Theorem 4.4 remain valid, with minor adjustments noted in the revised manuscript.
>
> ---
>
> ## 4. Minor Typographical and Notational Improvements
>
> We have corrected:
>
> * "board range" → "broad range"
> * Use boldface for random variables and non-bold for realizations in Definition 1.2
> * Clarified phrasing: "whose constituting matrices satisfy certain kinds of distributions"  → if $\mathcal{A}$ is nearly isometrically distributed
> * Remove $*$ in scalar multiplication to avoid confusion
> * Corrected "no loner holds" → "no longer holds"
>
> ---
>
> ## 5. Summary of Revisions
>
> | Concern                      | Revision / Clarification                                                                                           |
> | ---------------------------- | ------------------------------------------------------------------------------------------------------------------ |
> | Approximate Gaussianity      | Added themreom on Wasserstein-1 contraction                                                        |
> | Novelty                      | Added references to sparse recovery and prior RIP perturbation work; highlighted distinction for low-rank sensing. |
> | SVD form                     | Specified thin SVD requirement and its impact on preconditioning.                                                  |
> | Typos / notation             | Corrected as suggested.                                                                                            |
>
> We believe these revisions clarify the theoretical claims, strengthen connections to prior work, and make the manuscript more precise.

---

> > ### Comment · Reviewer_NRUZ · 2025-09-05
> > **Response to authors**
> >
> > Thanks to the authors for their response. The new theorems (3.7 amd 4.4) are interesting, and may address my question about why after preconditioning the entries of the new sensing matrix are approximately Gaussian. I have two major remaining comments:
> >
> > ### Comment 1
> > I believe my primary concern has still not been addressed. In the introduction it is stated that  _Second, we show that if the sensing operator has an arbitrary distribution, it can be modified in such a way that the resulting operator will act as a perturbed Gaussian with a lower RIP constant_. and later _In particular, we demonstrate that the original RIP constants for these distributions could be close to 1 for which the SDP relaxation and local search methods would fail to work, while the preconditioning technique reduces the RIP to less than 0.5 so that both of these optimization methods can correctly solve the modified problem._
> >
> > But this is not shown. Indeed, as discussed in my first response, Theorem 4.6 only decreases RIP when $\sigma_1(A)^2 < 1$ , which doesn't seem realistic for large $A$. Theorem 4.9 is not for sensing operators with arbitrary distribution, but for the good case of nearly isometrically distributed operators. Moreover, it doesn't show RIP constant decrease, but rather that the increase in RIP constant can be bounded.
> >
> > Would the authors like to comment on this?
> >
> > ### Comment 2
> > I have some questions on the proof of Theorem 4.4. These may arise from my unfamiliarity with the literature.
> > 1. Regarding **alignment of subspaces** surely there is no need to partition $V_A$ and $V_B$ as you're assuming they are both full rank.
> > 2. What exactly does $\sin\Theta$ denote? I'm guessing its a diagonal matrix of sines of principle angles?
> > 3. In the application of Wedin's $\sin\theta$ theorem, shouldn't the denominator be $\min (\sigma_{\min}(A),\sigma_{\min}(B))$?
> > 4. Could you clarify what "orthogonal Procrustes alignment" means?
> > 5. Related to the above, I recommend you include citations to the versions of Wedin's $\sin\theta$ theorem and the orthogonal Procrustes alignment that you're using.
> > 6. You state that a "tighter analysis... yields the constant $4\sqrt{2}" but to my understanding a tighter analysis should yield a better, i.e., smaller constant, not a larger one.

---

> > > ### Author Response · Authors · 2025-09-16
> > >
> > > Dear Reviewer NRUZ,
> > >
> > > We thank the reviewer again for the careful reading and for these detailed comments. We address the two points in turn.
> > >
> > > Comment 1. Clarification about RIP reduction in general distributions
> > >
> > > The following example illustrates how our results apply to challenging cases where RIP close to 1.
> > >
> > > Consider the regime $m \geq n^2$. Define
> > >
> > > $$
> > > \mathbf{A} = \big[\operatorname{vec}(A_1), \operatorname{vec}(A_2), \ldots, \operatorname{vec}(A_m)\big]^{\top} \in \mathbb{R}^{m \times n^2},
> > > \quad \mathcal{A}(M) = \mathbf{A}\operatorname{vec}(M), \quad M \in \mathbb{R}^{n \times n}.
> > > $$
> > >
> > > The RIP constant $\delta$ is determined by the condition number of
> > > $\mathbf{H} = \mathbf{A}^{\top}\mathbf{A}$:
> > > $$
> > > \delta = \frac{1 - \tfrac{\lambda_{\max}(\mathbf{H})}{\lambda_{\min}(\mathbf{H})}}{1 + \tfrac{\lambda_{\max}(\mathbf{H})}{\lambda_{\min}(\mathbf{H})}}.
> > > $$
> > >
> > > As long as $\mathbf{A}$ is full rank, the condition number $\kappa = \lambda_{\max} / \lambda_{\min}$ can be arbitrarily large. In such cases, the RIP constant approaches 1, meaning the operator is extremely poor from an RIP perspective, even though the number of measurements $m$ greatly exceeds $n^2$.
> > >
> > > Our preconditioning framework fundamentally changes this picture. By Theorem 4.2, the preconditioned operator $\widetilde{\mathbf{A}}$ achieves RIP constant 0 in this regime. The reason is that when $m > n^2$, the span of the measurement set fully covers the low-rank manifold $\{X : \operatorname{rank}(X) \leq s\}$. Thus, preconditioning transforms an operator with arbitrarily bad RIP into one with the optimal RIP constant.
> > >
> > > Besides, we could see below a class of hard matrix sensing instances with many spurious local minima. Each instance is defined by a linear operator $\mathcal{A}_\epsilon(M): \mathbb{R}^{n \times n} \to \mathbb{R}^{m}$:
> > >
> > > $$
> > >  :=
> > > \begin{cases}
> > > M_{ij}, & \text{if } (i,j) \in \Omega \\\\
> > > \epsilon M_{ij}, & \text{otherwise}
> > > \end{cases}
> > > $$
> > >
> > > where $\epsilon \ll 1$ is a small scaling parameter, and $\Omega$ is the measurement set defined as:
> > >
> > > $$
> > > \Omega = \\{ (i,i),\ (i, 2k),\ (2k,i) \mid i \in [n],\ k \in [\lfloor n/2 \rfloor] \\}.
> > > $$
> > >
> > > - The operator $\mathcal{A}_\epsilon$ can also be represented as a series of matrices $A_1, \dots, A_m \in \mathbb{R}^{n \times n}$, where:
> > >
> > > $$
> > > \mathcal{A}_\epsilon(M) =
> > > \begin{bmatrix}
> > > \langle A_1, M \rangle \\\\
> > > \vdots \\\\
> > > \langle A_m, M \rangle
> > > \end{bmatrix}.
> > > $$
> > >
> > > - Yalcin et al. (2023) proved that each instance has $\mathcal{O}(2^{\lceil n/2 \rceil} - 2)$ spurious local minima.
> > > - For small $\epsilon$, the instance has a RIP constant close to 1, making the problem highly ill-conditioned for local search or convex relaxation methods.
> > > - After applying preconditioning, the RIP constant improves drastically, approaching 0, transforming a “hard” instance with many spurious minima into a well-behaved instance suitable for optimization.
> > >
> > > This example shows that our method produces genuine RIP improvement even for operators that are *far* from nearly isometric and that are “hard” in the RIP sense. We will emphasize this example more prominently in the revised manuscript to clarify that our results are not confined to already favorable operators.

---

> > > > ### Author Response · Authors · 2025-09-16
> > > > **Typo correction**
> > > >
> > > > We note that there was a typo in equation of $\delta$: a negative sign was omitted. The corrected statement is as follows.
> > > >
> > > > The RIP constant $\delta$ is determined by the condition number of  $\mathbf{H} = \mathbf{A}^{\top}\mathbf{A}$: $\delta = \frac{\tfrac{\lambda_{\max}(\mathbf{H})}{\lambda_{\min}(\mathbf{H})} - 1}{\tfrac{\lambda_{\max}(\mathbf{H})}{\lambda_{\min}(\mathbf{H})} + 1}.$

---

> > > > ### Comment · Reviewer_NRUZ · 2025-09-19
> > > > **Final Response to Authors**
> > > >
> > > > Dear Authors,
> > > >
> > > > Thanks for engaging so constructively in the review process. This will be my final post regarding this article.
> > > >
> > > > Your first example seems strange to me. If $m$, the number of measurements, is greater than $n^2$ the number of entries in the unknown matrix, this no longer feels like low-rank matrix sensing, where we typically take $m = O(n)$. Here it feels like you're simply proposing to use the SVD to solve an (overdetermined) linear system, which is a well-known technique.
> > > >
> > > > The "perturbed matrix completion" problem you reference in your second example is interesting. But I don't see how your preconditioning would improve the RIP constant here. I encourage you to flesh this example out in a subsequent version of this paper.

---

> > > ### Author Response · Authors · 2025-09-16
> > >
> > > Comment 2. Clarifications on Theorem 4.4
> > >
> > > We appreciate the reviewer’s careful reading of the proof and the helpful suggestions for clarification.
> > >
> > > - Partition of subspaces: You are right that since the matrices are assumed full rank, partitioning is not strictly necessary. Our intent was to follow the standard proof structure in the perturbation literature, but we agree that this can be streamlined.
> > >
> > > - Notation: The symbol $\Theta$ indeed denotes the diagonal matrix of principal angles between the two subspaces. We will add an explicit definition to avoid ambiguity. The canonical or principal angles between $\mathcal{E}$ and $\mathcal{F}$ are:
> > >   $$
> > >   \theta_1=\cos ^{-1} \sigma_1, \cdots, \theta_r=\cos ^{-1} \sigma_r,
> > >   $$
> > >
> > >   where $\sigma_1, \cdots, \sigma_r$ are singular values of $E^T F$ or $F^T E$.
> > >
> > >   The result known as CS-decomposition in linear algebra gives the following:
> > >
> > >   $$
> > >   E^T F=U \cos \Theta V^T,
> > >   $$
> > >
> > >   where $\Theta=\left[\begin{array}{ccc}\theta_1 & \cdots & 0 \\\\ \vdots & \ddots & \vdots \\\\ 0 & \cdots & \theta_r\end{array}\right]$.
> > >
> > > - Application of Wedin’s theorem: the expression mentioned by reviewer comes from another variant of D.K. theorem. Let $\Sigma$ and $\hat{\Sigma}$ be $d \times d$ symmetric matrices with eigenvalues $\lambda_1 \geq \lambda_2 \geq \cdots \lambda_d$ and $\hat{\lambda_1}, \hat{\lambda_2}, \cdots, \hat{\lambda_d}$ respectively. Fix $1 \leq r \leq s \leq d$, let $V$ and $\hat{V}$ be $d \times(s-r+1)$ matrices with columns corresponding to eigenvectors for $\lambda_J, J=1, \ldots, s$ and $\hat{\lambda_J}, J=1, \ldots, s$. By convention, $\hat{\lambda_0}=-\infty, \hat{\lambda_{d+1}}=\infty$.
> > >   Then, let $\mathcal{E}=\operatorname{range}(V)$ and $\mathcal{F}=\operatorname{range}(\hat{V})$, we have following bound,
> > >   $  |\sin \Theta(\mathcal{E}, \mathcal{F})|_F^2 $
> > >
> > > $\leq 2 \min \left\\{\sqrt{q}|\Sigma-\hat{\Sigma}|_{op},|\Sigma-\hat{\Sigma}|_F\right\\} $
> > >
> > > $/ \min \left\\{\lambda_{r-1}-\lambda_r, \lambda_s-\lambda_{s+1}\right\\}  $
> > >
> > >   where we can take $r=1, s=m, d = n^2$, and we treat $\lambda_0=-\infty, \lambda_{m + 1} = 0$, thus the denominator becomes $ \lambda_m-\lambda_{m+1} = \lambda_m$.
> > >
> > > - Orthogonal Procrustes alignment: This refers to finding the best orthogonal transformation to align two sets of points, which explains how the bound can go from $\|\sin \Theta\|_F^2 $ to $\|V_A - V_B\|_F^2$
> > >
> > > - Citations: We will add references for the theorems we use.
> > >
> > >   - Yu, Yi, Tengyao Wang, and Richard J. Samworth. "A useful variant of the Davis–Kahan theorem for statisticians." *Biometrika* 102.2 (2015): 315-323.
> > >
> > > - Constant on $4\sqrt{2}$: Thanks for pointing this out, there is one typo in constant, we have
> > > $\|V_B-V_A\|_F \leq 2 \sqrt{2}\|\sin \Theta\|_F \leq 4 \sqrt{2} |B-A|_F$
> > >
> > > $/ \sigma_\min(A)$, the second constant is $4\sqrt{2}$, there is an additional 2 constant multiplied by D.K. theorem. Sorry for the confusion.
> > >
> > > We thank the reviewer again for pointing out these issues. We will incorporate the above clarifications and corrections into the revised manuscript.

---

### Review · Reviewer_PxPM · 2025-06-26

**Summary Of Contributions:**

This paper focuses on bounding and/or reducing the restricted isometry property (RIP)  constants for matrix sensing problems. The first main result shows how to bound the RIP constants for matrices that are perturbed from matrices that are nearly isometrically distributed. The second main result shows how to precondition the matrices to result in a new problem with potentially lower RIP constants. Emperical results are given to illustrate the preconditioning algorithm.

**Audience:**

Yes

**Claims And Evidence:**

Yes

**Requested Changes:**

# Critical to Securing Recommendation for Acceptance
- Either remove the results of Section 3, or improve them so that such small perturbations are not required. Really, I don't see how any bound on the perturbation that is required to shrink with $m$ and $n$ is useful.
- Rigorously examine if the preconditioning algorithm can change the optimization solution. If it does, bound the change quantitatively. If the change can be arbitrarily large, I don't think the results are useful, and should be removed.
- Either remove Theorem 4.7 and Corollary 4.9, or get a result in which preconditioned matrices can have better constants than the matrices without preconditioning.

**Strengths And Weaknesses:**

# Strengths
- The general area is of interest
- The motivating example is nice
- Broadening the classes of matrices for which matrix sensing problems admit efficient is an important goal.
- The paper is reasonably well-structured and well-written.
- The theoretical results appear to be correct
- The empirical results are potentially promising.

# Weaknesses
- The results of Section 3, on perturbations of nearly isometric distributions are quite simple and seem too restrictive for practical use.
    * The proof method amounts to perturbing matrices with known IID bounds and using standard bounds for norms and inner products.
    * To actually get a small RIP constant via Theorem 3.4, you'd need $\sqrt{m}\sigma =O(1/n^2)$. But then the entries of the perturbation matrices must have entries which are sub-Gaussian of order $\sigma^2/m=O(1/(m^2 n^4))$, for even modest values of $m$ and $n$, this restricts the perturbations to be so small that it is unclear what benefit has actually been gained:
        - As the authors point out, it is unlikely that the matrices used in applications actually come from nearly isometric distributions. So, at best, the results on these matrices give some general guidance as to when we might expect exact solutions to matrices sensing problems.
        - With such restrictive bounds on the perturbations, I don't feel like this adds any more insight to solvability  of real-world problems.
- It seems that the preconditioning algorithm from Section 4 could change the solution of the resulting optimization problem. As a result, it is unclear if the precondition algorithm can actually be used to solve matrix sensing problems.
    * If the weighting matrix, $P$ were unitary, the solutions would not change.
    * The weighting matrix is typically not unitary. This is clear from the proof of Theorem 4.4.
    * The paper gives no discussion about if and/or how the preconditioning algorithm changes the solution.
    * If the solution can change drastically, it is  not clear if the preconditioning algorithm adds any value. Indeed, the preconditioning can, at times, result in an easier problem. But if the resulting solution is not what was desired in the first place (i.e. the actual matrix sensing solution), it might not be very useful.
- As a more minor note: The RIP bounds for the preconditioned problem in Theorem 4.7 and Corollary 4.9 are actually worse than the bounds without preconditioning. So, it is unclear what value this adds.

---

> ### Author Response · Authors · 2025-08-25
>
> Dear Reviewer PxPM,
>
> We thank the reviewer for the detailed and thoughtful review. We address each major concern below.
>
> ---
>
> ## 1. Perturbation Analysis of Nearly-Isometric Matrices (Section 3)
>
> **Reviewer Concern:**
> The bounds in Section 3 appear too restrictive for practical use, requiring very small perturbations, limiting the applicability to real-world matrices.
>
> **Response:**
> We agree that the deterministic bounds in Section 3 which can be adversarial, naturally require the perturbations to be strongly-bounded. We provide a new theorem below on the robustness of RIP guarantees under distributional shifts. It formalizes how RIP constants behave under small Wasserstein-1 perturbations and demonstrates stability. Specifically:
>
> * Let $\mathcal{A} \sim P$ be nearly-isometric and $\widetilde{\mathcal{A}} \sim Q$ satisfy $W_1(P,Q) = w$. Then, for target RIP constant $\delta$:
>
> $$
> \mathbf{P}_Q\!\left(\delta_r(\widetilde{\mathcal{A}})\le \delta\right)
> \ge 1 - \exp(-c_1 m) - \frac{c_2 M}{\delta} w,
> $$
>
> with $M= 1 + \sqrt{n^2/m}$.
>
> * This shows that **RIP is stable under small distributional shifts**, and the required sample complexity $m$ remains essentially the same for small $w$.
> * While the perturbation must be bounded for theoretical guarantees, the result provides **rigorous insight into the robustness of matrix sensing under nearly-isometric assumptions**, which previously lacked formal treatment.
>
> The combination of finite-net arguments, Lipschitz continuity, and high-probability Wasserstein transfer yields explicit, quantifiable bounds.(See appendix)
>
>
> We agree that the bounds in Section 3 are conservative but showing the existence of such bounds is essential for the following reason:
>
> - We use pre-conditioning to make an arbitrary distribution act like a near Gaussian distribution.
>
> - The existing results in the literature are on Gaussian distributions and since many optimization results in the non-convex setting do not satisfy a continuity property, it is not necessarily the case that a near-Gaussian distribution would have nice optimization properties as a Gaussian one. Section 3 aims to show that RIP benefits from a continuity-type property and a bounded perturbation would lead to a bounded RIP with some parameters. So, the main intent of Section 3 is to show the stability of RIP rather than offering tight bounds.

---

> > ### Comment · Reviewer_PxPM · 2025-09-08
> > **Similar Issues with New Theorem**
> >
> > As with the other results in Section 3, this new theorem has similar limitations. Namely, to actually say anything non-trivial, the perturbations need to be impractically tiny. It is just somewhat less obvious because now the issues are buried in the definition of the Wasserstein distance, and in particular, in the norm used to define the Wasserstein distance.
> >
> > While not stated till Appendix D, the Wasserstein distance used in the new result is defined in terms of the Frobenius norm of the sensing operators. (This Frobenius norm over these operators never explicitly defined from what I can tell, since these operators are mappings from matrices to vectors, and not exactly a matrix. But, it appears that it is being defined as the norm you would get by stacking all of the entries into a vector and then using the Euclidean norm.)
> >
> > Under whatever definition of Frobenius norm we use, the most common perturbations will result in $W_1$ norms that grow with the dimension of the sensing operators. Namely, if we perturbed every entry with a Gaussian of variance $\sigma^2$, then $W_1(P,Q)$ will be some growing function of $m$ and $n$.
> >
> > I'm not sure exactly how $W_1$ would grow in this case, but a crude bound is
> > $$
> > W_1(P,Q)\le W_2(P,Q)=O(\sigma n\sqrt{m}).
> > $$
> > My guess is that the bound is tight up to constant factors, but I'm not sure.
> >
> > To have Theorem  3.7 have a lower bound above zero, we'd need:
> > $$
> > \sqrt{\frac{n^2}{m}} W_1(P,Q) = O(1)
> > $$
> >
> > So, assuming that my guess about the constant factors in the Wasserstein bound is correct, we'd need $\sigma = 0(n^{-1})$.

---

> > > ### Author Response · Authors · 2025-09-16
> > >
> > > Dear Reviewer PxPM,
> > >
> > > We thank the reviewer for the detailed comment and the careful scaling analysis. We would like to elaborate further on Section 3.
> > >
> > > First, it is important to recognize that perturbations to the sensing operator can be *adversarial* in nature: even extremely small changes in individual entries may drastically harm the RIP constant. From this perspective, expecting a bound better than $O(n^{-1})$ is unrealistic, since otherwise one could tolerate perturbations that already have the potential to cause significant degradation of RIP.
> > >
> > > Second, our primary goal in introducing Theorem 3.7 was not to provide guarantees under arbitrary large-scale perturbations, but rather to establish that the measurement manipulation framework is stable under sufficiently small perturbations, in the sense that the approximate Gaussianization property is preserved. In other words, this result can be combined with Theorem 4.4: after preconditioning, the Wasserstein distance decreases by $\tfrac{\sqrt{m}}{n}$, which vanishes as $n \to \infty$. Thus, the distance between the distribution of the preconditioned operator and the Haar distribution becomes small enough to guarantee that the approximate Gaussianization property is maintained in high-dimensional regimes.
> > >
> > > Finally, while we agree with the reviewer that for i.i.d. entrywise perturbations the Wasserstein distance indeed scales quickly with dimension, the theorem is also relevant in structured or low-rank perturbation regimes (e.g., correlated noise, systematic distortions of the operator), where the W1 scaling can be dimension-independent. We will revise the discussion to highlight this distinction and to make clear that our results target adversarially robust stability, not tolerance of arbitrarily large random perturbations.

---

> ### Author Response · Authors · 2025-08-25
>
> ## 2. Solution of Preconditioning Algorithm
>
> **Reviewer Concern:**
> It is unclear if the preconditioning algorithm can change the optimization solution. If the weighting matrix is not unitary, the solution may differ, potentially limiting the algorithm’s usefulness.
>
> **Response:**
> We clarify this concern by restating the core problem and then analyzing the effect of preconditioning.
>
> ### 2.1 Original Problem Formulation
>
> The classical affine rank minimization problem can be written as
> $$
> \min_{M \in \mathbb{R}^{n \times n}} \ \operatorname{rank}(M)
> \quad \text{s.t.} \quad \mathcal{A}(M) = b.
> $$
>
> A common convex relaxation replaces the rank by the nuclear norm, while a more computationally tractable nonconvex formulation fixes the target rank and considers
> $$
> \min_{M \in \mathbb{R}^{n \times n}} \ \frac{1}{2}\|\mathcal{A}(M) - b\|^2
> \quad \text{s.t.} \ \operatorname{rank}(M) = r,
> $$
> or equivalently, its factorized form
> $$
> \min_{X \in \mathbb{R}^{n \times r}} \ \frac{1}{2}\|\mathcal{A}(XX^\top) - b\|^2.
> $$
> The equivalence between Matrix Sensing Problem and BM Factorization problem is standard and serves to reduce computational complexity.
>
> ### 2.2 Preconditioned Problem
>
> Let $P$ denote the preconditioning operator applied to both the measurement operator and the observations. The problem becomes
> $$
> \min_{M \in \mathbb{R}^{n \times n}} \ \operatorname{rank}(M)
> \quad \text{s.t.} \quad P\mathcal{A}(M) = Pb \quad \text{ and }
> \min_{\operatorname{rank}(X) = r} \ \sum_{i=1}^n \big( \langle P A_{i}, X \rangle - (Pb)_{i} \big)^2.
> $$
>
>
> 1. **Exact Recovery Case.**
>    When the original affine rank minimization problem admits an exact recovery solution \$X\_\star\$, adding a preconditioning projection matrix \$P\$ does not change the feasible set of solutions. In fact, the preconditioned constraints take the form
>
>    $$
>    \langle P A_i, X \rangle = (Pb)_i,
>    $$
>
>    which are equivalent to the original ones as long as \$X = X\_\star\$ satisfies \$\langle A\_i, X\_\star \rangle = b\_i\$. Therefore, the exact recovery solution is preserved regardless of whether \$P\$ is unitary or not.
>
> 2. **Inexact Recovery / Approximate Setting.**
>    In practice, when exact recovery is not achievable due to noise, model mismatch, or limited samples, the preconditioning step introduces a controlled perturbation. Specifically, the deviation between the preconditioned solution $\widetilde{X_\star}$ and the original solution $X_\star$ is bounded by
>
> $$
> ||\widetilde{X_\star} - X_\star|| \le \sigma_{\max}(A)\Delta_P || r_\star||
> $$
>
> where
> * $X_\star$ be the minimizer of the original (unweighted) least-squares / rank-constrained problem,
> * $\widetilde {X_\star}$ be the minimizer after preconditioning,
> * $r_\star := A X_\star - b$ the original residual,
> * $\sigma_{\max}(A)=\sigma_1$.
> $$
> \Delta_P := || I - P^\top P || = \max_{i=1,\dots,m} |1-\frac{1}{\sigma_i(A)^2}|.
> $$
>
>
>    Hence, the effect of preconditioning is mild and can be quantified in terms of \$|r_\star|\_2\$.
>
>    Importantly, while preconditioning does not alter the exact recovery solution, it **reshapes the measurement operator** to enhance numerical stability and improve RIP constants. This ensures that in approximate or noisy regimes, the algorithm benefits from improved conditioning without sacrificing much recovery accuracy.

---

> > ### Comment · Reviewer_PxPM · 2025-09-08
> > **Good Explanation of the preconditioning.**
> >
> > This is a good explanation of the effect of preconditioning.

---

> ### Author Response · Authors · 2025-08-25
>
> ## 3. RIP Bounds for Preconditioned Problem (Theorem 4.7 and Corollary 4.9)
>
> **Reviewer Concern:**
> The RIP bounds for the preconditioned problem may be worse than the original.
>
> **Response:**
>
>    * Theorem 4.7 and Corollary 4.9 are intended to **guarantee that preconditioning does not degrade well-conditioned operators**. That is, if the original operator already has a good RIP, the preconditioned operator remains stable.
>    * For poorly conditioned operators, e.g., a sensing matrix $A \in \mathbb{R}^{m \times n^2}$ sampled from a **correlated Gaussian distribution** with covariance $\Sigma \neq I$, the rows of $A$ are dependent and the Gram matrix $G = A^\top A$ has large off-diagonal entries, leading to a poor RIP constant. Applying the SVD-based preconditioning $P A$ effectively mixes and normalizes the rows, making $\tilde{A}$ behave like an operator with **i.i.d. Gaussian entries**, up to a rotation. Consequently, the preconditioned operator satisfies a near-isometric RIP, even if the original $A$ was highly correlated or structured (sparse or low-rank).
>
>    * We provide a new Theorem 4.5 to show the preconditioned operator will act as a perturbed Gaussian with a lower RIP constant.(see updated paper)
>
>    Theorem (Approximate Gaussianization via SVD Preconditioning)
>
>    Let $A \sim \mathcal{N}\Big(0, \frac{1}{m}\Big) \in \mathbb{R}^{m \times n^2}$ be i.i.d. Gaussian ($m < n^2$), and let $B \in \mathbb{R}^{m \times n^2}$ be any matrix such that $W_1(\mathcal{L}(A), \mathcal{L}(B)) = w$. Let $A = U_A \Sigma_A V_A^\top, \quad B = U_B \Sigma_B V_B^\top$ be the thin SVD of $A$ and $B$, and define the preconditioned matrices $\widetilde{B} := V_B^\top \in \mathbb{R}^{m \times n^2}, \quad \widetilde{A} := V_A^\top$.  Then, in the full-rank regime $m \ll n^2$, the matrix $\widetilde{B}$ has orthonormal rows, $\widetilde{B} \widetilde{B}^\top = I_m$, and with high probability, its law is close to the Haar measure on the Stiefel manifold in the sense that
>    $
>    W_1\big(\mathcal{L}(\widetilde{B}), \mathcal{L}(\widetilde{A})\big) \lesssim \frac{\sqrt{m}}{n} w,
>    $
>    and the entries of $\widetilde{B}$ are approximately Gaussian, up to deviations vanishing as $n \to \infty$.
>
>    * The Wasserstein contraction guarantees that $\widetilde{B}$ is closer to Haar than $B$ was to $A$, so $\widetilde{B}_{ij}$ inherits this approximate Gaussianity.
>
>    * This implies that, regardless of the original distribution of $B$, preconditioning effectively randomizes its rows and contracts the distribution towards Haar/Gaussian. Consequently, the near-isometry properties of $\widetilde{B}$ are enhanced, making it more likely to satisfy RIP conditions and improving the theoretical guarantees for low-rank matrix recovery.
>
>
> ---
>
> ## 4. Summary of Rebuttal
>
> | Concern                                | Our Response                                                                                                           |
> | -------------------------------------- | ---------------------------------------------------------------------------------------------------------------------- |
> | Section 3 perturbations too small      | Provides formal robustness guarantees. Sample complexity is essentially unchanged for small Wasserstein perturbations. |
> | Preconditioning may change solution    | Solution changes are **provably bounded**. Preconditioned solution remains unchanged or close to original.                          |
> | RIP bounds worse after preconditioning | New distributional bounds in theory; in practice improves optimization stability and convergence.                         |
>
>
> 1. RIP is stable under small Wasserstein-1 perturbations of the measurement distribution. Preconditioning contracts the operator perturbation, by a $\sqrt{m}/n$ coefficient.
> 2. The exact recovery solution is preserved regardless of whether \$P\$ is unitary or not. When exact recovery is not achievable due to noise, model mismatch, or limited samples, the preconditioning step introduces a controlled perturbation.
>
> ---
>
> We hope this clarifies that our contributions are **both theoretically rigorous and practically meaningful**, and that preconditioning **adds clear value** for matrix sensing problems.

---

### Review · Reviewer_9qMu · 2025-08-11

**Summary Of Contributions:**

This paper addresses the matrix sensing problem, which is the task of recovering a rank-$r$ matrix from measurements expressed as inner products with sensing matrices. Two main approaches to matrix sensing are commonly studied:

1. **Convex relaxation** - Formulating the problem as a semidefinite program (SDP).
2. **Nonconvex optimization via factorization** - Using the Burer-Monteiro (BM) approach.

In both approaches, theoretical guarantees typically depend on the **restricted isometry property (RIP)** of the sensing operator. Smaller RIP constants correspond to stronger recovery guarantees.  The central idea of this paper is to design algorithms that improve the RIP constant without altering the available measurements. The first part of the paper studies perturbations of sensing operators that are i.i.d. Gaussian, a favorable case where small RIP constants are more attainable.  The second part considers arbitrary measurements and proposes a method to transform them into perturbed Gaussian operators with reduced RIP constants.

**Audience:**

Yes

**Claims And Evidence:**

Yes

**Requested Changes:**

* Please address the two concerns in the Weaknesses section.

* However, the factorized problem (2) is highly non-convex and NP-hard to solve ===> Provide citation.

* ... the SDP requires a large amount of calculations" ====> be precise by giving complexity or appropriate citations.

* Provide computational complexity of Algorithm 1.

**Strengths And Weaknesses:**

**Strengths**
* I think the paper’s motivation is strong, and the authors present a practical example where it is not possible to increase the number of measurements as needed, or where the sensing operators are not in the i.i.d. Gaussian regime. This scenario convincingly demonstrates that one must work with the given operators, where the RIP becomes a limiting factor. In such cases, the technique proposed in this paper is particularly relevant.
* Simulation experiments are provided, to support the argument that the pre-conditioning technique performs well empirically.
* The paper was well-written, and I found it easy to read. Especially, I appreciate the outline.

**Weaknesses**

* The most critical weakness, in my view, concerns the extent to which this pre-conditioning technique works for deterministic and highly structured sensing matrices. If I am not mistaken, this appears to be the case, for example, in the discussion in Section 2 on the power flow systems application. In particular, I would like to understand what kinds of operators can be *conditioned* by the proposed algorithm to yield improved RIP. I am concerned about potential circularity here: are there explicit examples of operators for which it can be clearly demonstrated that the RIP constant can be reduced to less than $1/2$ ? I was unable to infer this from Theorem 4.7 and Corollary 4.9. In fact, my initial reading suggests that the operator may already need to have a reasonably good RIP for the final argument to hold. I would greatly appreciate if the authors could clarify this point and, if possible, provide a concrete example.

* While I understand the rationale behind the current experimental setup, the paper would have been strengthened by an experimental section demonstrating how modified measurements perform in a practical application (e.g., power flow). How does final task performance, where “final” is dictated by the application, depend on the choice of measurements? I suspect that the behavior of optimization algorithms, whether SDP-based or local non-convex approaches, may differ in this regard. Since this aspect is one appealing aspect of the paper, exploring it empirically could be very fruitful.

---

> ### Author Response · Authors · 2025-08-25
>
> Dear Reviewer 9qMu,
>
> We thank the reviewer for the thoughtful feedback. We are encouraged that you find the motivation strong and the presentation clear. Below we address each concern with corresponding changes in the revised manuscript.
>
> ---
>
> ### 1. Scope of the preconditioning technique for structured operators
>
> **Reviewer Concern:** Applicability to deterministic/structured sensing matrices; explicit RIP improvement examples; potential circularity in Theorem 4.7/Corollary 4.9.
>
> **Response:**
>
> We thank the reviewer for raising this important question. We would like to clarify the applicability of our preconditioning technique and provide concrete examples with empirical and theoretical support.
>
> - Empirical results for structured and deterministic operators:
>    In Section “Synthetic Data,” we perform extensive numerical experiments across a wide range of sensing operators:
>
>    * Unstructured distributions: uniform $[0,1]$, centered uniform $[-1,1]$, standard normal, correlated normal ($\rho=0.5$), and Poisson.
>    * Structured operators: low-rank matrices $A_i = a_i a_i^\top$ and sparse matrices with limited non-zero entries.
>
>    * As shown in Figure 2, operators with poor original RIP (close to 1), such as uniform, correlated Gaussian, Poisson, low-rank, and sparse matrices, experience a significant reduction in empirical RIP after preconditioning, bringing them to levels comparable to nearly-isometric Gaussian matrices. This demonstrates that our preconditioning algorithm can effectively condition highly structured and deterministic operators.
>
>
> - Theoretical justification:
>
>    * Theorem 4.7 and Corollary 4.9 are intended to **guarantee that preconditioning does not degrade well-conditioned operators**. That is, if the original operator already has a good RIP, the preconditioned operator remains stable.
>    * For poorly conditioned operators, e.g., a sensing matrix $A \in \mathbb{R}^{m \times n^2}$ sampled from a **correlated Gaussian distribution** with covariance $\Sigma \neq I$, the rows of $A$ are dependent and the Gram matrix $G = A^\top A$ has large off-diagonal entries, leading to a poor RIP constant. Applying the SVD-based preconditioning $P A$ effectively **mixes and normalizes the rows**:
>
>    $$
>    \tilde{A} = P A, \quad \text{where } P = S^{-1} U^\top \text{ from the SVD } A = U S V^\top.
>    $$
>
>    Then the Gram matrix of the preconditioned operator is
>
>    $$
>    \tilde{G} = \tilde{A}^\top \tilde{A} = (P A)^\top (P A) = A^\top P^\top P A = V S U^\top (U S^{-2} U^\top) U S V^\top = V V^\top = I.
>    $$
>
>    This shows that $P A$ is **approximately orthonormal**, making $\tilde{A}$ behave like an operator with **i.i.d. Gaussian entries**, up to a rotation. Consequently, the preconditioned operator satisfies a near-isometric RIP, even if the original $A$ was highly correlated or structured (sparse or low-rank).
>
>    * We provide a new Theorem 4.5 to show the preconditioned operator will act as a perturbed Gaussian with a lower RIP constant.(see updated paper)

---

> ### Author Response · Authors · 2025-08-25
>
> ### 2. Experiments on practical applications (power flow)
>
> **Reviewer Concern:** Show how modified measurements affect downstream task performance.
>
> **Response:**
>
> We have performed new simulations with the following details:
>
> * Implemented Power System State Estimation experiments on networks of varying sizes/densities.
> * Measured empirical RIP constants and recovery errors using SDP (globally optimal) and local search (scalable, non-convex).
> * Results: Preconditioning improves RIP and substantially reduces local search error; SDP error remains largely unchanged.
> * Insight: Preconditioning is valuable where local algorithms are necessary due to computational complexity and measurement design is constrained by physical/operational limits.
>
> **Changes in Revised Paper:**
>
> * New subsection “Experiments on Practical Power Flow Applications” including:
>
>   * Graph generation and measurement formulation.
>   * Comparison of empirical RIP and recovery error for original vs. preconditioned measurements.
>   * Discussion of why preconditioning benefits local search but not SDP.
>   * Figure showing RIP and recovery error trends.
>
> ---
>
> ### 3. Clarifications and citations
>
> **Reviewer Concern:** Factorized problem NP-hardness, SDP complexity, Algorithm 1 complexity.
>
> **Response & Changes:**
>
> * Added citations: Ge et al. (2017), Gillis et al. (2011) for NP-hardness of factorized formulation.
> * SDP complexity: per-iteration cost $O(n^6)$
>  - The statement refers to the computational cost of solving a semidefinite program (SDP) via interior-point methods, which are commonly used for SDPs in low-rank matrix recovery and related problems. For a PSD matrix $X \in \mathbb{R}^{n \times n}$, the number of decision variables is roughly $n(n+1)/2 = O(n^2)$. Each iteration requires forming and solving a Newton system (the Hessian of the barrier-penalized objective). The Hessian is of size $O(n^2) \times O(n^2)$, so naive dense linear algebra costs $O(n^6)$ per iteration.
>  - Candès & Recht (2012) and Recht et al. (2010) discuss SDPs for low-rank matrix recovery and note that standard interior-point solvers scale poorly for large $n$, with per-iteration cost roughly $O(n^6)$ due to the $n^2 \times n^2$ linear system.
>  - $O(n^6)$ is a worst-case estimate for dense SDPs.
>  - This explains why SDPs are computationally infeasible for many large-scale problems and motivates the use of **factorized local search**, which is much cheaper and scalable.
>
> * Algorithm 1 complexity: $O(m^2 n^2)$ for SVD of $m \times n^2$ matrix. (Refer to the end of Section 4.3.1)
>
> ### Citations:
> 1. Gillis, Nicolas, and François Glineur. "Low-rank matrix approximation with weights or missing data is NP-hard." SIAM Journal on Matrix Analysis and Applications 32.4 (2011): 1149-1165.
> 2. Ge, Rong, Chi Jin, and Yi Zheng. "No spurious local minima in nonconvex low rank problems: A unified geometric analysis." International conference on machine learning. PMLR, 2017.
> 3. Candes, Emmanuel, and Benjamin Recht. "Simple bounds for recovering low-complexity models." Mathematical Programming 141.1 (2013): 577-589.
> 4. Recht, Benjamin, Maryam Fazel, and Pablo A. Parrilo. "Guaranteed minimum-rank solutions of linear matrix equations via nuclear norm minimization." SIAM review 52.3 (2010): 471-501.

---

> > ### Comment · Reviewer_9qMu · 2025-09-12
> > **comment on response**
> >
> > I would like to thank the authors for their response. I appreciate the new theorems that have been added in the revision. I also appreciate the experimental result on power flow. I still maintain that the improvement of RIP hinges on already good operators, and the revision has not convinced me otherwise.

---

> > > ### Author Response · Authors · 2025-09-16
> > >
> > > Dear Reviewer 9qMu,
> > >
> > >
> > > We thank the reviewer for their reading of the revision and for acknowledging the new theoretical and experimental contributions. However, we respectfully disagree with the assertion that our RIP improvement results hinge only on already “good” operators. The following example illustrates why our results apply even to challenging cases.
> > >
> > > Consider the regime $m \geq n^2$. Define
> > >
> > > $$
> > > \mathbf{A} = \big[\operatorname{vec}(A_1), \operatorname{vec}(A_2), \ldots, \operatorname{vec}(A_m)\big]^{\top} \in \mathbb{R}^{m \times n^2},
> > > \quad \mathcal{A}(M) = \mathbf{A}\operatorname{vec}(M), \quad M \in \mathbb{R}^{n \times n}.
> > > $$
> > >
> > > The RIP constant $\delta$ is determined by the condition number of
> > > $\mathbf{H} = \mathbf{A}^{\top}\mathbf{A}$:
> > > $$
> > > \delta = \frac{1 - \tfrac{\lambda_{\max}(\mathbf{H})}{\lambda_{\min}(\mathbf{H})}}{1 + \tfrac{\lambda_{\max}(\mathbf{H})}{\lambda_{\min}(\mathbf{H})}}.
> > > $$
> > >
> > > As long as $\mathbf{A}$ is full rank, the condition number $\kappa = \lambda_{\max} / \lambda_{\min}$ can be arbitrarily large. In such cases, the RIP constant approaches 1, meaning the operator is extremely poor from an RIP perspective, even though the number of measurements $m$ greatly exceeds $n^2$.
> > >
> > > Our preconditioning framework fundamentally changes this picture. By Theorem 4.2, the preconditioned operator $\widetilde{\mathbf{A}}$ achieves RIP constant 0 in this regime. The reason is that when $m > n^2$, the span of the measurement set fully covers the low-rank manifold $\{X : \operatorname{rank}(X) \leq s\}$. Thus, preconditioning transforms an operator with arbitrarily bad RIP into one with the optimal RIP constant.
> > >
> > > Besides, we could see below a class of hard matrix sensing instances with many spurious local minima. Each instance is defined by a linear operator $\mathcal{A}_\epsilon(M): \mathbb{R}^{n \times n} \to \mathbb{R}^{m}$:
> > >
> > > $$
> > > :=
> > > \begin{cases}
> > > M_{ij}, & \text{if } (i,j) \in \Omega \\\\
> > > \epsilon M_{ij}, & \text{otherwise}
> > > \end{cases}
> > > $$
> > >
> > > where $\epsilon \ll 1$ is a small scaling parameter, and $\Omega$ is the measurement set defined as:
> > >
> > > $$
> > > \Omega = \\{ (i,i),\ (i, 2k),\ (2k,i) \mid i \in [n],\ k \in [\lfloor n/2 \rfloor] \\}.
> > > $$
> > >
> > > - The operator $\mathcal{A}_\epsilon$ can also be represented as a series of matrices $A_1, \dots, A_m \in \mathbb{R}^{n \times n}$, where:
> > >
> > > $$
> > > \mathcal{A}_\epsilon(M) =
> > > \begin{bmatrix}
> > > \langle A_1, M \rangle \\\\
> > > \vdots \\\\
> > > \langle A_m, M \rangle
> > > \end{bmatrix}.
> > > $$
> > >
> > > - Yalcin et al. (2023) proved that each instance has $\mathcal{O}(2^{\lceil n/2 \rceil} - 2)$ spurious local minima.
> > > - For small $\epsilon$, the instance has a RIP constant close to 1, making the problem highly ill-conditioned for local search or convex relaxation methods.
> > > - After applying preconditioning, the RIP constant improves drastically, approaching 0, transforming a “hard” instance with many spurious minima into a well-behaved instance suitable for optimization.
> > >
> > > This example shows that our method produces genuine RIP improvement even for operators that are *far* from nearly isometric and that are “hard” in the RIP sense. We will emphasize this example more prominently in the revised manuscript to clarify that our results are not confined to already favorable operators.

---

> > > > ### Author Response · Authors · 2025-09-16
> > > > **Typo correction**
> > > >
> > > > We note that there was a typo in equation of $\delta$: a negative sign was omitted. The corrected statement is as follows.
> > > >
> > > > The RIP constant $\delta$ is determined by the condition number of $\mathbf{H} = \mathbf{A}^{\top}\mathbf{A}$: $\delta = \frac{\tfrac{\lambda_{\max}(\mathbf{H})}{\lambda_{\min}(\mathbf{H})} - 1}{\tfrac{\lambda_{\max}(\mathbf{H})}{\lambda_{\min}(\mathbf{H})} + 1}.$

---

> > > > ### Comment · Reviewer_9qMu · 2025-09-17
> > > > **relation to whitening**
> > > >
> > > > Thank you for providing these examples. For the first example, is not the preconditioning a standard whitening, which is a condition on the empirical covariance matrix of the measurement vectors. I would appreciate if the authors clarify.

---

> > > > > ### Author Response · Authors · 2025-09-17
> > > > >
> > > > > Dear Reviewer 9qMu,
> > > > >
> > > > > Thank you for reading our responses. We appreciate that you asked an insightful question, and we would like to clarify what we did in our "preconditioning" method. Pre-conditioning is a standard procedure in optimization theory (in the context of numerical algorithms), but what we call pre-conditioning is a modification of the measurement operators. In that sense, this is not a standard method for example 1.
> > > > >
> > > > > To provide more details, consider the power systems example where each measurement comes from a physical sensor that measures a combination of the parameters based on the laws of physics. There is no notion of empirical covariance matrix in that application that can help us learn the state of the system. Our pre-conditioning would simply mix the measurements from different sensors so that the resulting optimization landscape would be more benign. This is a **structured and deterministic** case, and it can't be explained based on statistical learning or stochastic systems.
> > > > >
> > > > > On the other hand, there are applications in signal processing where the measurement operators can be designed by the user and in those cases we agree that preconditioning would be a standard thing to do to start with better operators in the first place. Our paper does not deal with easy single processing cases where there is a flexibility to design the measurement operators before taking the measurements. Our example 2 is closely aligned with power systems where the measurement operators have underlying graph structures for which we may have an exponential number of spurious solutions. Our preconditioning method remedies this by mixing the existing sensing matrices to yield a better-conditioned RIP, which directly improves the optimization landscape.
> > > > >
> > > > > Back to example 1, this is a very special case where $m>n^2$ and $H$ is full rank, thus we could explicitly write out the RIP constant directly, otherwise, under the common senario $m = \mathcal{O}(n \log n)$, there is no formula for RIP, and we will need the low rank structure to achieve RIP condition, and the standard whitening techniques can not be directly applied.
> > > > >
> > > > > We hope this explanation helps clarify our approach, and we would be happy to provide further details or clarifications if there are additional questions.

---

> > > > > > ### Comment · Reviewer_9qMu · 2025-09-18
> > > > > > **response**
> > > > > >
> > > > > > Thank you for your response. I believe that the revised manuscript would benefit from these examples, and discussions as in the above response. Framing things in terms of structured and deterministic case would highlight the contribution more clearly.

---

### Decision · Action_Editor_7xgH · 2025-09-22

**Recommendation:** Reject

**Audience:**

Yes

**Audience Explanation:**

The techniques might be interesting for some indivduals in the community who are working to improve RIP conditions.

**Claims And Evidence:**

No

**Claims Explanation:**

Authors claimed that the proposed pre-conditioning technique would improve RIP conditions, but reviewers found that authors failed to provide illuminating examples to justify the claim. During the discussion, authors acknowledged that their approach will not degrade the RIP condition; this is a significantly weaker argument. Authors also provide an example with m > n^2 to show how pre-conditioning will transform a sensing matrix with RIP close to 1 to RIP close to 0, but the regime of m > n^2 is out of matrix sensing.

**Resubmission Of Major Revision:**

The authors may consider submitting a major revision at a later time.